# Tumor heterogeneity and clonal cooperation influence the immune selection of IFN-γ-signaling mutant cancer cells

Jason B. Williams[1,5], Shuyin Li[1,5], Emily F. Higgs[1], Alexandra Cabanov[1], Xiaozhong Wang[2], Haochu Huang[1] & Thomas F. Gajewski[1,3,4]*

PD-1/PD-L1 blockade can promote robust tumor regression yet secondary resistance often occurs as immune selective pressure drives outgrowth of resistant tumor clones. Here using a genome-wide CRISPR screen in B16.SIY melanoma cells, we confirm *Ifngr2* and *Jak1* as important genes conferring sensitivity to T cell-mediated killing in vitro. However, when implanted into mice, these *Ifngr2*- and *Jak1*-deficient tumors paradoxically are better controlled immunologically. This phenotype maps to defective PD-L1 upregulation on mutant tumor cells, which improves anti-tumor efficacy of CD8[+] T cells. To reconcile these observations with clinical reports of anti-PD-1 resistance linked to emergence of IFN-γ signaling mutants, we show that when mixed with wild-type tumor cells, IFN-γ-insensitive tumor cells indeed grow out, which depends upon PD-L1 expression by wild-type cells. Our results illustrate the complexity of functions for IFN-γ in anti-tumor immunity and demonstrate that intratumor heterogeneity and clonal cooperation can contribute to immunotherapy resistance.

[1] Department of Pathology, The University of Chicago, Chicago, IL 60637, United States. [2] Department of Molecular Biosciences, Northwestern University, Evanston, IL 60208, United States. [3] Departments of Medicine, Section of Hematology/Oncology, Chicago, IL 60208, United States. [4] The Ben May Department for Cancer Research, The University of Chicago, Chicago, IL 60637, United States. [5]These authors contributed equally: Jason B. Williams, Shuyin Li. *email: tgajewsk@medicine.bsd.uchicago.edu

Immune checkpoint blockade therapy targeting the negative regulatory receptors CTLA-4 and/or PD-1 has transformed cancer care, being FDA approved in at least 14 distinct cancer entities[1–6]. However, despite these successes, many patients do not respond clinically, and some patients initially show clinical tumor regression yet subsequently progress with therapeutic resistance. The mechanisms of primary resistance are only beginning to be understood, and include tumor-cell-intrinsic oncogenic alterations, parallel immune suppressive pathways, adaptive resistance mechanisms, and the composition of the commensal microbiota that set the overall immune system tone[7–12]. Secondary resistance arises under strong immune selective pressure and is also beginning to become characterized, with evidence for the emergence of tumor cells that lose antigen expression, or are deficient for expression of class I MHC or antigen processing machinery[12–14]. Recent investigations uncovered deficiencies in IFNγR signaling in resistant tumor cell variants derived from patients who progressed after initially responding to anti-PD-1 therapy[15–17]. This latter phenomenon is of particular interest, as the functional roles of IFN-γ in antitumor immunity are complex and include both positive and negative regulatory activities. IFN-γ induces upregulation of class I MHC and of antigen processing molecules, exerts antiproliferative effects, and also promotes production of chemokines that have antiangiogenic properties in addition to promoting effector T cell recruitment[18,19]. However, IFN-γ also induces upregulation of key negative regulatory molecules, including PD-L1 and indolamine-2,3-dioxygenase (IDO), and additionally can support T cell apoptosis during immune responses in vivo[20–22]. It is thus conceivable that the functional consequences of loss of IFNγR signaling on tumor cells will depend on the net sum of complex activities of this cytokine in individual contexts, and whether positive versus negative immune regulatory effects are dominant in a given scenario.

Genome-wide screens using CRISPR/Cas9 genome editing have offered an additional tool for identifying candidate immune resistance factors through in vitro selection assays. Such studies have confirmed that disruption of the genes encoding β2 M, TAP1, and TAP2 lead to defective CD8+ T cell-mediated killing in vitro[23–25]. Mutations in the IFN-γ signaling pathway also have been identified in such screens, including disruptions in Jak1, Jak2, Ifngr1, and Ifngr2[24,25]. A novel functional importance of Ptpn2 in T cell-mediated tumor control also has been identified using a CRISPR screening approach[23].

Following the identification of a genetic disruption in tumor cells that results in diminished T-cell-mediated killing in vitro, it is critical to validate the functional consequence of that genetic alteration upon implantation of tumors into immunocompetent mice in vivo. In the process of a CRISPR/Cas9 genome editing screen in B16.SIY melanoma cells to discover gene alterations mediating resistance to CD8+ T cell killing in vitro, we confirmed the importance of IFNγR signaling on tumor cells in this process. However, when these clonal mutant tumor cells were implanted in vivo into mice, they paradoxically showed improved immune-mediated tumor control. This phenomenon mapped to defective upregulation of the immune-inhibitory ligand PD-L1 by tumor cells. Nonetheless, when a mixture of wild-type (WT) and IFNγR-mutant tumor cells was implanted in vivo to mimic the tumor heterogeneity often seen in patients, anti-PD-L1 therapy selected for outgrowth of the IFNγR2-mutant tumor cell clones. This phenomenon depended upon PD-L1 provided by the WT tumor cells, and also IFN-γ made by antitumor T cells. These results highlight the complexity of IFN-γ functions during the dynamics of an antitumor immune response in vivo, and indicate that both tumor heterogeneity and clonal cooperation can be involved in the immune selection process.

## Results

**A genome-wide CRISPR screen identifies resistant mutants.** To identify tumor-cell-intrinsic genes essential for CD8+ T cell killing we performed a genome-wide CRISPR/Cas9 deletion library screen in B16F10 melanoma cells expressing the model antigen SIY (SIYRYYGL) fused with DsRed (designated B16.SIY), under in vitro selection pressure by SIY-specific 2 C CD8+ TCR transgenic (Tg) cytotoxic T cells (Supplementary Fig. 1a). We utilized a pooled mouse single-guide RNA (sgRNA) library containing 87,897 sgRNA sequences targeting 19,150 mouse protein-coding genes[26]. B16.SIY cells were transduced with a genome-scale gRNA lentivirus. In vitro-primed CD8+ T cells isolated from 2 C/Rag2−/− TCR transgenic mice were co-cultured with transduced B16.SIY tumor cells. To control the selection pressure, we optimized the T cell killing assay to obtain 99% target cell lysis at a 2:1 2 C/Rag2−/− T cell-to-target ratio (Supplementary Fig. 1b). After overnight incubation, 2 C/Rag2−/− T cells were removed and the remaining resistant B16.SIY tumor cells were allowed to recover for 1 week. Genes targeted by sgRNAs were identified by sequencing the gRNA cassette from genomic DNA of resistant tumor cells. 33 genes were identified (Supplementary Table 1), including H2k1 (H2-Kb), the class I MHC molecule responsible for presenting the SIY peptide. Two of the most frequently recovered sgRNAs targeted genes involved in the IFN-γ signaling pathway, namely Ifngr2 and Jak1 (Supplementary Fig. 1c). We focused our attention on Ifngr2 and Jak1, as loss-of-function mutations or deletions in these genes have been identified in patients with spontaneous resistance to anti-PD-1 checkpoint blockade[15–17].

**Generation of IFN-γ-insensitive B16.SIY tumor cells.** To be able to examine the role of tumor-intrinsic IFN-γ signaling in the antitumor immune response in vivo, we generated IFNγR2- and Jak1-mutant B16.SIY tumor cell lines by single-cell cloning using the sgRNAs identified from the CRISPR/Cas9 screen. To minimize founder effects that can arise from single-cell cloning, we began by generating monogenetic founder B16.SIY tumor cell clones and selected one that behaved similar to polyclonal B16. SIY tumors in vivo. Several criteria were imposed to select an optimal founder population; these included similar SIY expression based on DsRed intensity, comparable tumor growth kinetics to polyclonal B16.SIY tumors in vivo, and normal efficacy response to anti-PD-L1 blockade in vivo (Supplementary Fig. 2a and b). To mutate IFNγR2 and Jak1, we generated lentiviral vectors encoding sgRNAs targeting IFNγR2 or Jak1 and a BFP reporter to select for transductants. As a control, the founder cell population was transduced with empty vector (EV-BFP). After transduction of sgRNAs we transfected cells with a Cas9-encoding vector and selected for Cas9-expressing cells with blasticidin. Single cells were sorted based on BFP expression and screened for loss of responsiveness to IFN-γ by failure to upregulate H-2Kb after IFN-γ stimulation in vitro (Supplementary Fig. 2c).

We used two different sgRNAs for both IFNγR2 and Jak1 to control for off-target effects. One clone was chosen for each sgRNA and IFN-γ-insensitivity was tested (Fig. 1a). To confirm loss-of-function mutations in both alleles the genotype was determined by sequencing (Fig. 1b). We found a premature stop codon in all alleles, with the exception of Jak1 Mutant 1 (Jak1 Mut.1) allele 2, which obtained an in-frame six base pair deletion (Fig. 1b). To validate our initial in vitro CRISPR/Cas9 screen we exposed IFNγR2- and Jak1-mutant tumor cells to activated 2 C/Rag2−/− T cells to evaluate lysis. Indeed, IFN-γ-insensitive tumor cells were relatively resistant to T cell-mediated killing in vitro, confirming our genome-wide screen results (Fig. 1c).

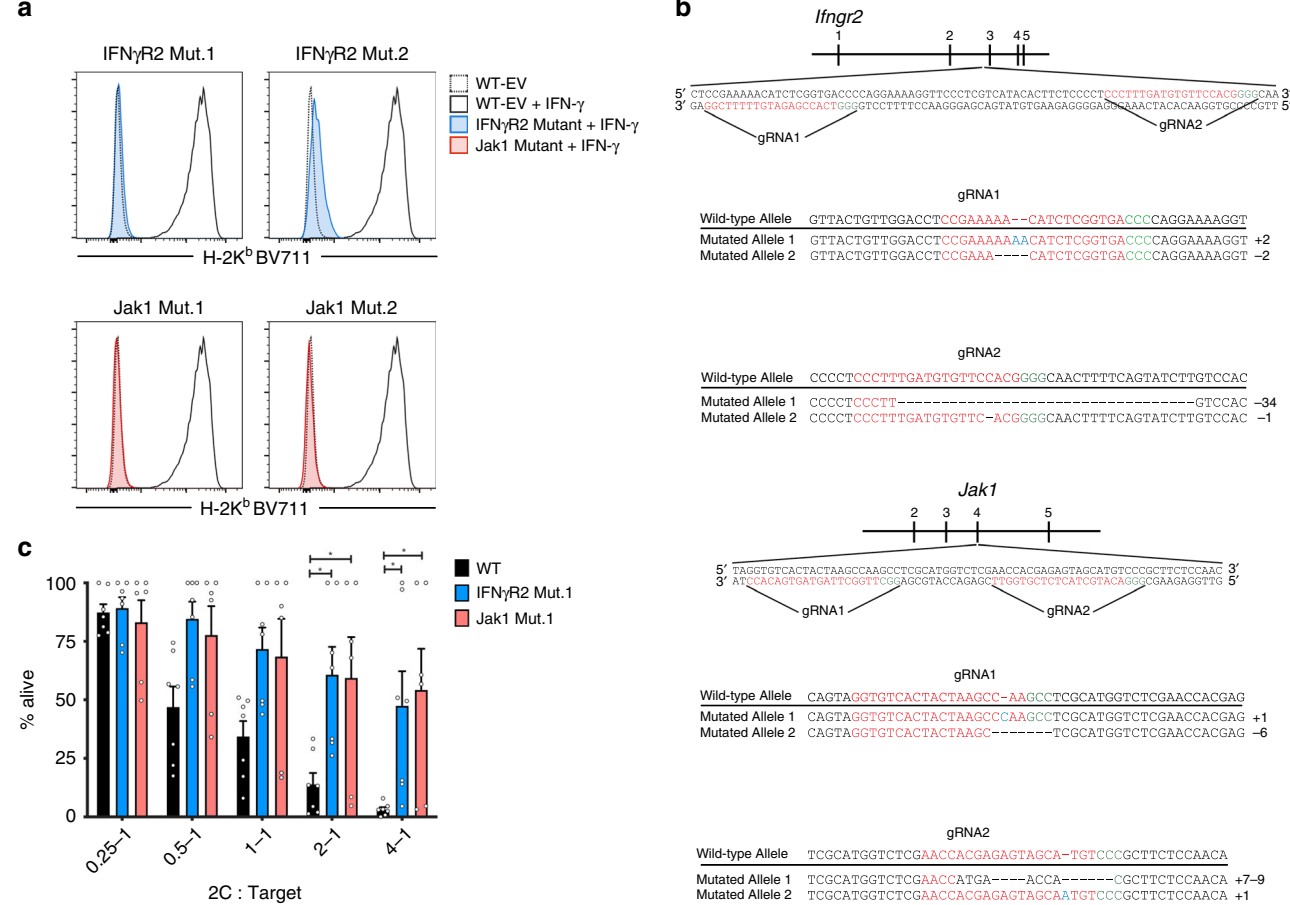

**Fig. 1 A genome-wide CRISPR/Cas9 screen identifies *Ifngr2* and *Jak1* as essential for T cell-mediated killing of B16.SIY cells in vitro. a** In vitro assessment for loss of IFN-γ signaling. Tumor cell clones were stimulated with 10 ng/mL IFN-γ for 16 h and measured for H-2K$^b$ upregulation by flow cytometry. **b** Genotype of IFNγR2- and Jak1-mutant cell lines. Colors highlight features as follows: red, gRNA sequence; green, PAM; blue, nucleotide insertion. **c** Relative resistance of IFNγR2- and Jak1-mutant tumor cells to T cell-mediated killing in vitro. Tumor cells were incubated with pre-primed 2 C T cells for 24 h and remaining cells were measured by live/dead staining. $n = 7$ assays; data are pooled from three independent experiments. Results are expressed as mean ± s.e.m. Statistical significance was determined by a two-way ANOVA Bonferroni post-hoc test (**c**). *$p < 0.05$.

**IFN-γ-insensitive tumors are spontaneously controlled in vivo**. In order to assess the behavior of IFNγR2- and Jak1-mutant tumor cells in vivo, we subcutaneously implanted mutant or WT tumor cells into C57BL/6 mice and tracked tumor growth. Unexpectedly, both IFNγR2- and Jak1-mutant B16.SIY tumors were spontaneously rejected by day 14. This was not due to increased immunogenicity from BFP expression as WT-BFP B16. SIY tumors grew progressively (Fig. 2a). To rule out the possibility that spontaneous tumor control was due to off-target effects from CRISPR/Cas9 mutagenesis, we retrovirally reintroduced IFNγR2 or Jak1 into IFNγR2- and Jak1-mutant cells, respectively, and confirmed successful IFN-γ signaling by upregulation of H-2K$^b$ after IFN-γ stimulation in vitro (Fig. 2b, c). When implanted into mice, tumors with restored IFN-γ signaling exhibited progressive growth kinetics comparable to WT tumors (Fig. 2d, e). The observed restoration of tumor growth kinetics indicates that no increase in immunogenicity due to on- or off-target mutations from CRISPR/Cas9 mutagenesis was responsible for the spontaneous control of IFN-γ-insensitive tumors.

To determine if increased tumor control was specific to the B16.SIY model system or if this phenomenon might be observed in other models, we assessed whether MC38 tumors similarly mutated exhibited delayed tumor growth. As with the generation of IFNγR2- and Jak1-mutant B16.SIY tumor cells, we generated MC38 IFNγR2- and Jak1-mutant tumors by establishing a

founder cell line and a similar single-cell CRISPR/Cas9 mutagenesis method. We confirmed the loss of IFN-γ signaling in IFNγR2- and Jak1-mutant MC38 tumor cell clones (Supplementary Fig. 3a). Similar to the B16.SIY model, IFN-γ-insensitive MC38 tumors exhibited slower tumor growth in vivo (Supplementary Fig. 3b), which was reversed after the reintroduction of IFNγR2 (Supplementary Fig. 3c and d). B16.SIY and MC38 were selected because they are known to be immunogenic; therefore, we tested whether this phenotype is observed in a less antigenic model. To this end, we mutated IFNγR2 and Jak1 in B16F10 tumor cells using the same single-cell method and confirmed successful disruption of IFN-γ signaling (Supplementary Fig. 3e). Because B16F10 tumors are highly aggressive, we utilized the standard number of 100,000 tumor cells to allow for more time for the immune system to mount a response. Consistent with our previous observations, IFNγR2- and Jak1-mutant B16F10 tumors also exhibited slower tumor growth, although the difference in growth rate was less pronounced presumably due to decreased antigenicity (Supplementary Fig. 3f). These data show that, in multiple tumor models, defective tumor cell-intrinsic IFN-γ signaling leads to slower tumor outgrowth in vivo.

**Improved tumor control is dependent on CD8$^+$ T cells.** To determine if the spontaneous tumor control was dependent upon

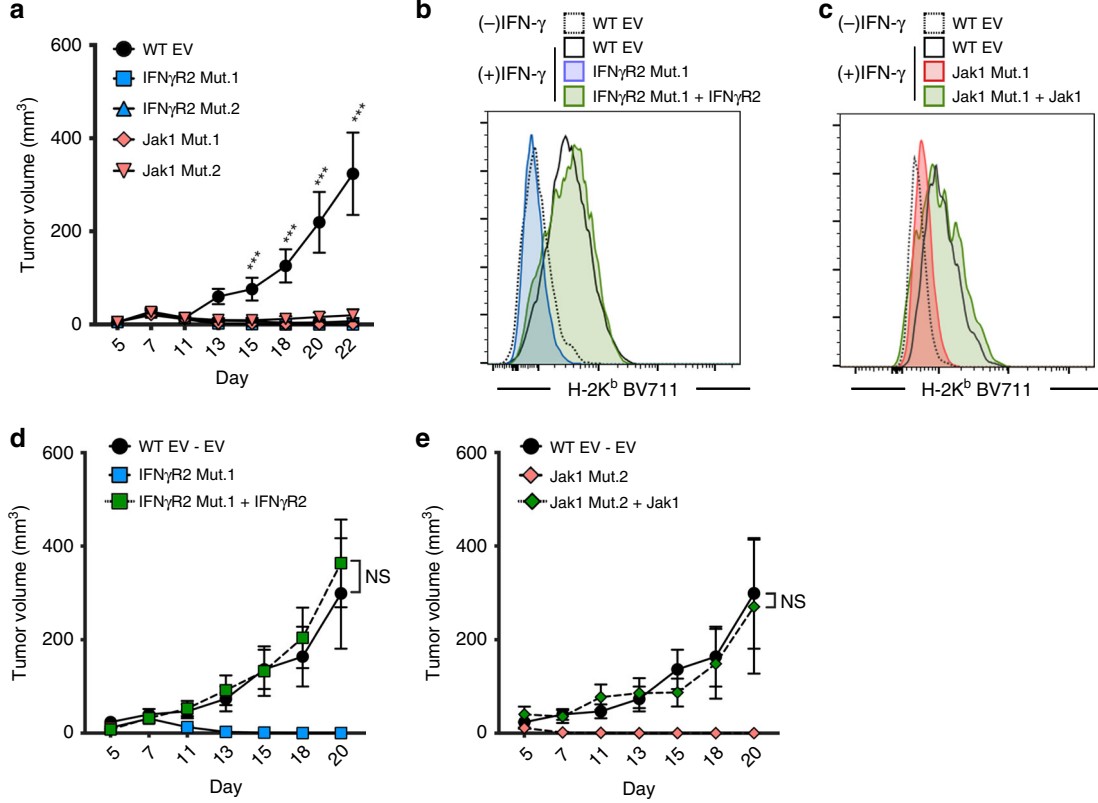

**Fig. 2 IFNγR2- and Jak1-mutant tumors are spontaneously controlled in vivo. a** Tumor growth curves of WT, IFNγR2-, and Jak1-mutant tumors. $n = 10$ mice; data are pooled from two independent experiments. **b, c** Representative histogram showing restored IFN-γ signaling after retroviral reintroduction of IFNγR2 (**b**) or Jak1 (**c**). Tumor cells were stimulated with 10 ng/mL IFN-γ for 16 h and measured for H-2K$^b$ upregulation by flow cytometry. $n = 4$ stimulations; data is representative of two independent experiments. **d, e** Restoration of progressive tumor growth of IFNγR2 Mut.1 tumors with reintroduced IFNγR2 (**d**) and Jak1 Mut.1 tumors with reintroduced Jak1 (**e**). $n = 15$ mice (WT EV, IFNγR2 Mut.1, and IFNγR2 Mut.1 + IFNγR2), $n = 10$ mice (Jak1 Mut.2), $n = 5$ mice (Jak1 Mut.2 + Jak1); data are pooled from three (**d**) or one (**e**) independent experiments. Results are expressed as mean ± s.e.m. Statistical significance was determined by a two-way ANOVA Bonferroni post-hoc test (**a, d, e**). *** $p < 0.001$.

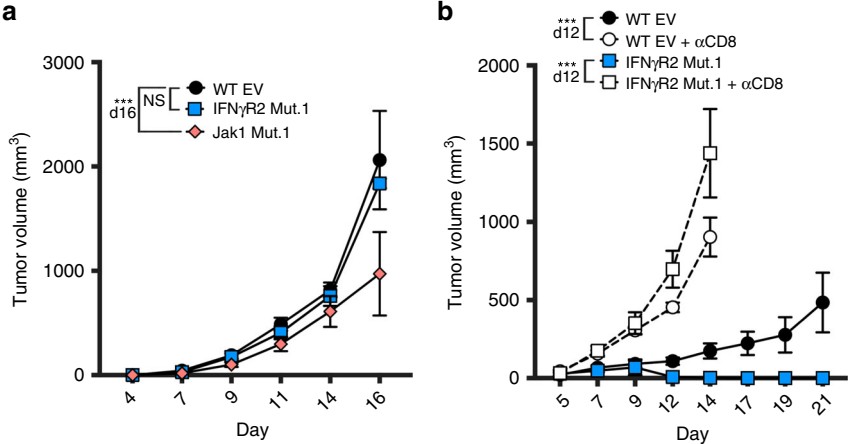

**Fig. 3 CD8$^+$ T cells are required for the spontaneous control of IFN-γ-insensitive tumors. a** Tumor outgrowth of WT, IFNγR2 Mut.1, and Jak1 Mut.2 tumors in Rag2$^{-/-}$ mice. $n = 6$ mice; data are pooled from two independent experiments. **b** Tumor outgrowth of WT, IFNγR2 Mut.1 CD8$^+$ T cell-depleted mice. Mice received 200 μg of anti-CD8α (clone YTS169.4). $n = 8$ mice (WT and IFNγR2 Mut.1) and $n = 7$ mice (Jak1 Mut.1); data are pooled from two independent experiments. Results are expressed as mean ± s.e.m. Statistical significance was determined by a two-way ANOVA Bonferroni post-hoc test (**a, b**). *** $p < 0.001$.

adaptive immunity, we inoculated Rag2$^{-/-}$ mice with WT, IFNγR2-, or Jak1-mutant tumor cells and tracked tumor growth. No difference in tumor growth between IFNγR2-mutant and WT tumors was observed, although Jak1-mutant tumor growth was slightly delayed (Fig. 3a). Knowing that tumor control was dependent on

the adaptive immune system we investigated which immune cell compartment was necessary for tumor control. B16 melanoma is known to express low levels of class I MHC at baseline, which is upregulated by IFN-γ. Therefore, it was conceivable that failure to upregulate MHCI molecules by IFN-γ-insensitive tumors could lead

to tumor control by NK cells. Alternatively, a low level of MHCI may be sufficient to escape destruction by NK cells, and the spontaneous tumor control was dependent on other immune cell subsets. Therefore, we depleted CD8$^+$ T cells, NK cells or both with depleting antibodies in C57BL/6 mice 1 day before and 5 and 10 days after engrafting WT or IFNγR2-mutant tumor cells. Successful depletion was confirmed 3 days after antibody administration (Supplementary Fig. 4). While tumor control was maintained in NK-depleted mice (Supplementary Fig. 5a), CD8-depleted mice failed to control IFNγR2-mutant tumors (Fig. 3b) and no synergy was observed when both NK and CD8$^+$ T cells were depleted (Supplementary Fig. 5b). These data indicate that CD8$^+$ T cells were required to control IFN-γ-insensitive tumors.

**The antitumor CD8$^+$ T cell response is augmented**. Because CD8$^+$ T cells were necessary for control of IFN-γ-insensitive tumors in vivo, we characterized the endogenous antitumor CD8$^+$ T cell response. The frequency of SIY-reactive CD8$^+$ T cells was evaluated in the spleen by an IFN-γ ELISPOT assay. On day 7 after tumor inoculation, we observed a modest increase in the frequency of IFN-γ-producing effector cells specific for the SIY antigen (Fig. 4a). We also measured the frequency of tumor-antigen-specific CD8$^+$ T cells within the tumor microenvironment by H-2K$^b$/SIY-pentamer staining. On day 7 after tumor challenge, a 3-fold increase in the frequency of H-2K$^b$/SIY$^+$ CD8$^+$ tumor infiltrating lymphocytes (TILs) was observed in IFNγR2- and Jak1-mutant tumors compared to WT tumors (Fig. 4b). To rule out the possibility that off-target mutations were responsible for the increase in H-2K$^b$/SIY$^+$ CD8$^+$ TIL we measured the frequency of antigen-specific CD8$^+$ TILs in IFNγR2- and Jak1-mutant tumors with reintroduced IFNγR2 or Jak1. Restoration of IFN-γ signaling in tumor cells reverted the frequency of antigen-specific CD8$^+$ TILs in IFN-γ-insensitive tumors to levels comparable to WT (Fig. 4c, d). Together, these data indicate that the absence of IFN-γ signaling in tumor cells resulted in increased accumulation of tumor-antigen-specific CD8$^+$ T cells in the tumor microenvironment.

**IFN-γ drives a tumor-intrinsic immune modulatory program**. Knowing that CD8$^+$ T cells were the mediator of tumor control we next sought to determine whether CD8$^+$ TILs exhibited improved effector functions in the context of IFNγR2- or Jak1-mutant tumors. However, CD8$^+$ TILs isolated on day 7 after tumor engraftment showed no difference in expression of *Ifng*, *Gzmb*, *Tnfa*, *Prf1*, and *Il2* by qRT-PCR in WT, IFNγR2-, and Jak1-mutant tumor contexts (Supplementary Fig. 6). These data suggest that the CD8$^+$ TIL compartment contains the necessary cytotoxic functions to eradicate IFNγR2- and Jak1-mutant tumors, indicating that an alteration on the tumor cell side may be responsible for the improved spontaneous tumor control observed.

To investigate whether IFN-γ-insensitive tumor cells showed decreased expression of a negative immune regulatory factor, RNASeq was performed on purified tumor cells from WT, IFNγR2-, and Jak1-mutant tumors on day 7 after tumor engraftment. Many of the differentially expressed genes found were shared between IFNγR2- and Jak1-mutant tumor cells (Fig. 5a and Supplementary Data 1). Overlapping downregulated genes included those involved in antigen presentation (*H2-K1*, *H2-Aa*, *H2-T23*, Tap1, *Tap2*, *B2m*), immune functions (*CD274*, *Serpinb9*, *Icam2*, *Cxcl9*, *Ccl5*, *Sema7a*), extracellular matrix (*Lgals3bp*, *Gjp2*, *Itgb3*), ubiquitination (*Trim21*, *Dtx3l*, *Herc6*), and GTPase activity (*Ligp1*, *Tgtp1*, *Gbp2*, *Gbp7*) (Fig. 5b, c). However, a key negative regulatory molecule was also expressed at lower levels, *CD274* (PD-L1). Indolamine-2,3-deoxygenase

(IDO), another known IFN-γ-induced negative immune regulatory gene[27], was minimally expressed by tumor cells and not different between conditions. Since total tumor digests have been shown to upregulate IDO in previous work[27], we isolated tumor cells and host APCs from tumors on day 7 and analyzed IDO expression by qRT-PCR. We found that tumor cells themselves expressed very little transcript for IDO whereas significant levels of IDO transcript were observed among the host APCs (Supplementary Fig. 7). These data point to a broad IFN-γ-induced genetic program induced in WT but not IFNγR2- or Jak1-mutant tumor cells early in the antitumor immune response, with most of these genes being positive factors for antitumor immunity, but the key negative regulator PD-L1 is also induced in WT tumor cells yet lost in IFNγR signaling mutants.

**Restored PD-L1 expression re-establishes tumor growth**. We hypothesized that one possibility to explain the spontaneous tumor control of IFN-γ-insensitive tumors was their failure to upregulate PD-L1, an important adaptive resistance mechanism. We first measured PD-L1 expression on the host and tumor compartments in WT, IFNγR2-, and Jak1-mutant tumors following implantation in vivo. We found a high level of PD-L1 expression on host APCs and on WT tumor cells; in contrast, IFN-γ-insensitive tumor cells showed minimal PD-L1 expression (Fig. 6a–c). This was confirmed by at the transcript level by qRT-PCR analysis on sorted tumor cells (Fig. 6d). These results suggested that reduced PD-L1 expression might be responsible for improved tumor control. To test this hypothesis, we retrovirally restored expression of PD-L1 in IFNγR2- and Jak1-mutant tumor cells (Fig. 6e). Restored expression of PD-L1 in IFNγR2- and Jak1-mutant tumor cells re-established the progressive growth kinetics comparable to WT tumors (Fig. 6f). These results suggested that PD-L1 may be acting directly at the T cell:tumor cell interface to inhibit CD8$^+$ T cell cytotoxicity. If these were true, by removing antigen-dependent tumor recognition by CD8$^+$ T cells, IFN-γ-insensitive tumors should no longer be spontaneously rejected. To test this, we deleted the H-2K$^b$ gene from the IFNγR2-mutant tumor cells by CRISPR/Cas9 mutagenesis and tracked tumor growth. Indeed, IFNγR2-mutant tumors lacking H-2K$^b$ grew progressively (Fig. 6g). These results indicate that IFNγR2- and Jak1-mutant tumors are better controlled immunologically through defective expression of PD-L1, in a manner dependent on direct tumor recognition by CD8$^+$ TILs.

**IFNγR2-mutants are selected when mixed with WT tumor cells**. Our data thus far implicate tumor-intrinsic IFN-γ signaling as a critical factor to blunt the initial T cell insult and to establish the immunosuppressive microenvironment through the upregulation of immune negative regulatory molecules, particularly PD-L1. In addition, PD-L1 expressed on host cells may partly compensate for dampening the antitumor immune response when PD-L1 is not expressed on IFN-γ-insensitive tumor cells. In recent reported cases of acquired immune resistance, tumors or patient-derived cell lines were found to contained loss-of-function mutations in components of the IFN-γ signaling pathway and tumor cells harboring these mutations increased in proportion after anti-PD-1 therapy[15–17]. Therefore, to investigate why IFN-γ-insensitive tumors were spontaneously controlled instead of being more aggressive, we reasoned that our mouse system differed in two major ways. First, the human tumor scenario almost certainly involves intratumoral heterogeneity, in which a minor subset of IFN-γ signaling-mutant tumor cell clones was selected out among a population of IFN-γ signaling competent tumor cell clones. Second, the clinical scenario also occurred in the context of anti-PD-1 antibody therapy, which

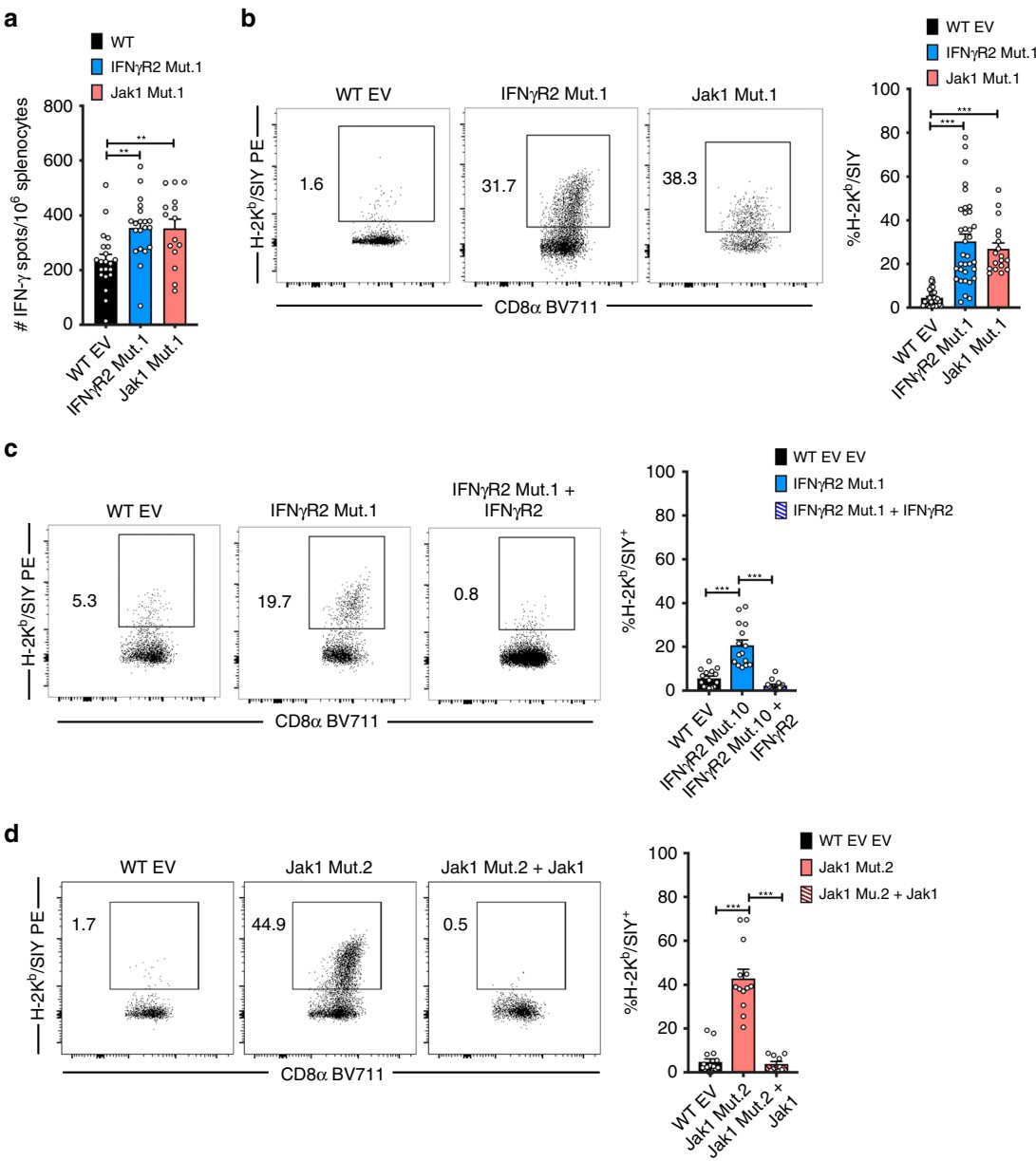

**Fig. 4 The antitumor CD8+ T cell response is augmented against IFN-γ-insensitive tumors. a** Frequency of splenic IFN-γ-secreting T cells in response to SIY measured by enzyme-linked immunospot (ELISPOT). Splenocytes were isolated from mice on day 7 after tumor engraftment and cultured with 100 μM SIY for 18 h. $n = 21$ mice (WT), $n = 20$ mice (IFNγR2 Mut.1), and $n = 10$ mice (Jak1 Mut.1); data are pooled from 4 (IFNγR2 Mut.1) or 2 (Jak1 Mut.1) independent experiments. **b** Representative flow cytometry plots (left) and cumulative data (right) of H-2K$^b$/SIY$^+$ CD8$^+$ TILs isolated on day 7 after engraftment of WT, IFNγR2 Mut.1, and Jak1 Mut.1 tumor cells. $n = 33$ mice (WT), $n = 34$ mice (IFNγR2 Mut.1), and $n = 18$ mice (Jak1 Mut.1); data are pooled from six (IFNγR2 Mut.1) or four (Jak1 Mut.1) independent experiments. **c** As in **b** but with IFNγR2 Mut.1 tumors with reintroduced IFNγR2. $n = 15$ mice (WT and IFNγR2 Mut.1), $n = 14$ mice (IFNγR2 Mut.1 + IFNγR2); data are pooled from three independent experiments. **d** As in **b** but with Jak1 Mut.1 tumors with reintroduced Jak1. $n = 17$ mice (WT), $n = 13$ mice (Jak1 Mut.1), and $n = 10$ mice (Jak1 Mut.1 + Jak1); data are pooled from three independent experiments. Results are expressed as mean ± s.e.m. Statistical significance was determined by a Kruskal-Wallis (non-parametric) test (**a–d**). *$p < 0.05$, **$p < 0.01$, ***$p < 0.001$.

would neutralize the negative effect of PD-L1 on the WT tumor cell clones. In theory, this would leave the antiproliferative and pro-immunogenic effects of IFN-γ to dominate, which would spare the IFN-γ signaling-mutants and allow their outgrowth. To test this hypothesis, we inoculated mice with a mixture of WT and IFNγR2-mutant tumor cells at a 1:1 ratio, and then treated the mice with anti-PD-L1 antibody. When tumors were measured over time, we observed slow but progressive tumor growth of the mixed tumor population, suggesting that the presence of WT

tumor cells provided missing negative regulatory signals (Fig. 7a). In approximately half (15/28) of mice with WT:IFNγR2-mutant mixed tumors that received anti-PD-L1 therapy, the tumor escaped. When these tumors were harvested and re-analyzed by flow cytometry, selection for the IFNγR2-mutant tumor cells was observed in 12/15 of these cases (Fig. 7b). To rule out the possibility that selection of IFNγR2-mutant tumor cells was due to factors independent of the immune system, Rag2$^{-/-}$ mice were inoculated with either mixed WT:WT or WT:IFNγR2-mutant

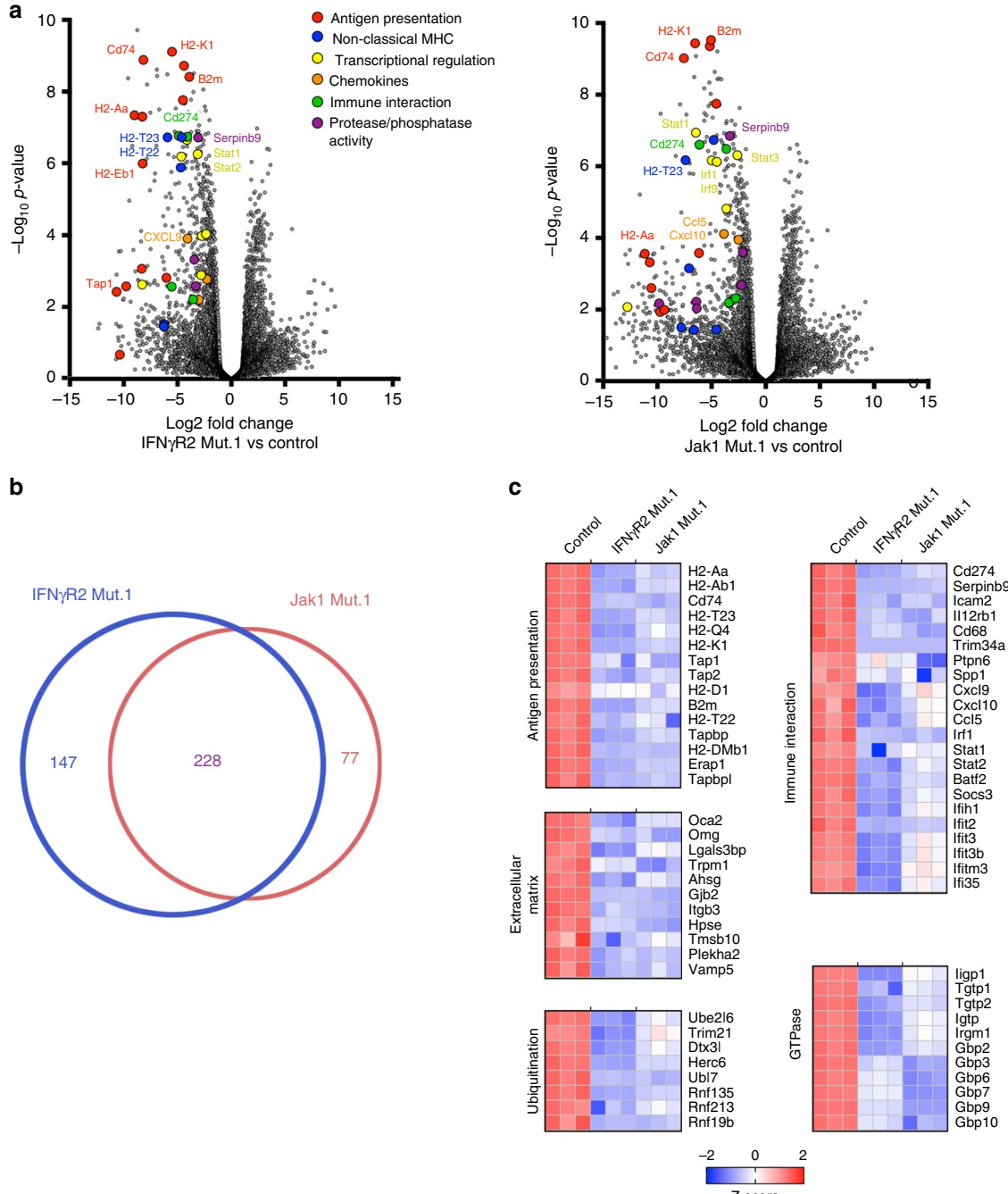

**Fig. 5 A complex genetic program is induced by IFN-γ signaling in tumor cells that includes PD-L1. a** Volcano plot of differentially expressed genes (DEGs) from IFNγR2- and Jak1-mutant tumor cells compared to WT tumor cells. Tumor cells were sorted on day 7 after tumor engraftment. **b** The number of unique and shared DEGs between IFNγR2- and Jak1-mutant tumor cells. **c** Selected downregulated genes in IFNγR2- or Jak1-mutant tumor cells compared to WT tumor cells grouped by biological pathway. Numerical values in heat map are expressed as Z-scores.

tumor cells, and no differences between the tumor clones was observed (Fig. 7c). We reasoned that this immune selection for IFNγR2 mutants would occur most robustly in the presence of a strong tumor-antigen-specific CD8$^+$ T cell response, as the selection mechanism would involve direct tumor cell recognition of antigen. We therefore analyzed the SIY-specific T cell response using H-2K$^b$/SIY-pentamer staining among the TILs. In fact, we found that in the 12 mice where selection of IFNγR2-mutant tumor cells occurred, the frequency of H-2K$^b$/SIY$^+$ CD8$^+$ TILs was greater compared to the three mice that grew out WT tumor cells (Fig. 7d, e). In fact, a greater frequency of H-2K$^b$/SIY$^+$

CD8$^+$ TILs correlated with a greater fraction of IFNγR2-mutant tumor cells emerging (Fig. 7f). Together, these data suggest that in the context of a strong tumor antigen-specific CD8$^+$ T cell response and PD-L1 blockade, IFN-γ-insensitive tumor cells have a selective growth advantage out of a mixture with WT tumor cells, allowing escape which equates to therapeutic resistance.

**IFN-γ from CD8$^+$ TILs drives selection for mutant tumor cells.** To address the mechanism behind the selection process of IFNγR2-mutant tumor cells, we hypothesized that IFN-γ-insensitive tumor cells could escape the direct antitumor effects of

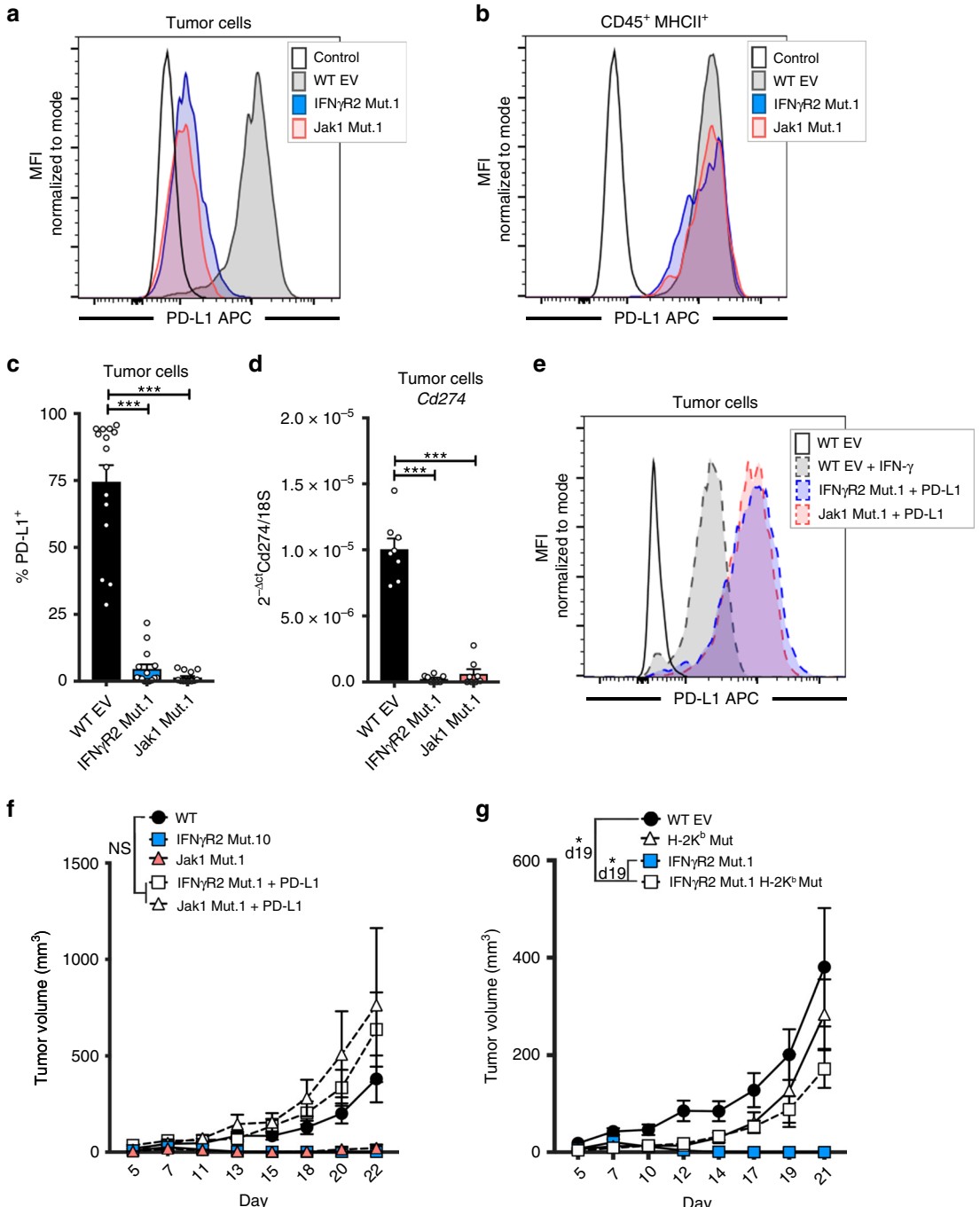

**Fig. 6 Re-establishment of PD-L1 restores progress tumor growth of IFN-γ-insensitive tumors. a–c** Representative histograms of PD-L1 expression on tumor cells (**a**) or CD45+ MHCII+ APCs (**b**) from IFNγR2 Mut.1, Jak1 Mut.1, or WT tumors and summary of percent PD-L1+ tumor cells (**c**). Analysis was performed on day 7 after tumor engraftment. $n = 15$ mice; data are pooled from three independent experiments. **d** mRNA expression of PD-L1 from sorted tumor cells on day 7 after tumor engraftment. $n = 8$ mice; data are pooled from two independent experiments. **e** Representative histogram of PD-L1 expression after retroviral introduction of PD-L1 in IFNγR2- and Jak1-mutant tumor cells in vitro. Expression of PD-L1 was compared to WT tumor cells with or without IFN-γ stimulation. **f** Tumor outgrowth curves when PD-L1 was re-expressed in IFNγR2- and Jak1-mutant tumor cells. $n = 10$ mice (WT, IFNγR2 Mut1, Jak1 Mut1, and IFNγR2 Mut.1 + PD-L1) and $n = 5$ mice (Jak1 Mut.1 + PD-L1); data are pooled from two independent experiments (IFNγR2) or from one experiment (Jak1). **g** Tumor outgrowth curves when H-2Kb was deleted in IFNγR2-mutant tumor cells. $n = 10$ mice (WT and IFNγR2 Mut.1), $n = 15$ mice (IFNγR2 Mut.1 H-2Kb Mut), and $n = 5$ mice (H-2Kb Mut); data are pooled from two independent experiments. Results are expressed as mean ± s.e.m. Statistical significance was determined by a Kruskal-Wallis (non-parametric) test (**b, c**) and a two-way ANOVA Bonferroni post-hoc test (**f, g**). *$p < 0.05$, ***$p < 0.001$.

IFN-γ, which would be preserved in tumor cells that have intact IFN-γ-signaling. Under this premise, we reasoned that the relative intratumoral quantity of IFN-γ would correlate with the selection of IFNγR2-mutant tumor cells. To test this, we inoculated mice with mixed WT:IFNγR2-mutant tumors and once tumors emerged they were excised. The tumor was split into two fractions, one fraction was mashed and assessed for IFN-γ content by ELISA within the interstitial fluid, and the other was used to measure the frequency IFNγR2-mutant and WT tumor cells. We found that the concentration of intratumoral IFN-γ positively

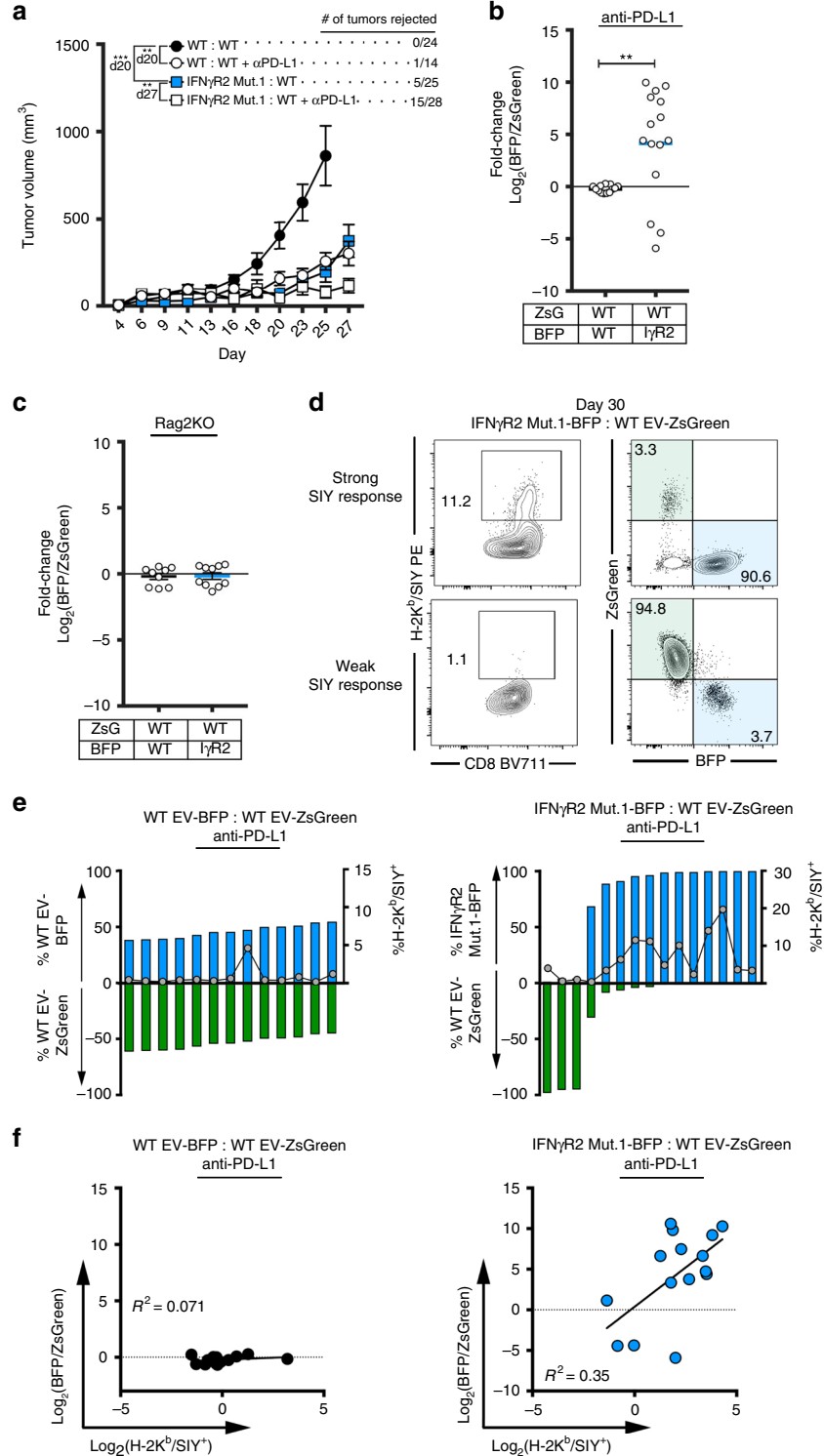

**Fig. 7 When mixed with WT tumor cells, IFN-γ-insensitive tumor cells selectively grow out following to anti-PD-L1 therapy in vivo. a** Tumor outgrowth of mixed WT and IFNγR2-mutant tumors with and without anti-PD-L1 therapy. $n = 24$ mice (WT:WT), $n = 14$ mice (WT:WT + anti-PD-L1), $n = 25$ mice (IFNγR2 Mut.1:WT), and $n = 28$ (WT:IFNγR2 Mut.1 + anti-PD-L1); data are pooled from four independent experiments. **b** Log$_2$(BFP/ZsGreen) fold-change after anti-PD-L1 therapy. **c** Fold-change of mixed tumors in Rag2$^{-/-}$ mice. $n = 9$ mice (WT:WT) and $n = 10$ mice (WT:IFNγR2 Mut.1); data are pooled from two independent experiments. **d** Representative flow plots of tumor composition of mixed WT:IFNγR2-mutant tumors in the context of a strong and weak immune response. **e** Tumor composition of mixed tumors from individual mice treated with anti-PD-L1. **f** Correlation between H-2K$^b$/SIY$^+$ CD8$^+$ TILs and selection for WT-BFP or IFNγR2 Mut.1 BFP. For (**b, e, f**) $n = 13$ mice (WT:WT + anti-PD-L1) and $n = 15$ mice (WT:IFNγR2 Mut.1 + anti-PD-L1); data are pooled from five independent experiments. Results are expressed as mean ± s.e.m. Statistical significance was determined by a two-way ANOVA Bonferroni post-hoc test (**a**) and a Kruskal-Wallis (non-parametric) test (**b, c**). Least squares regression was performed in (**f**). **$p < 0.01$, ***$p < 0.001$.

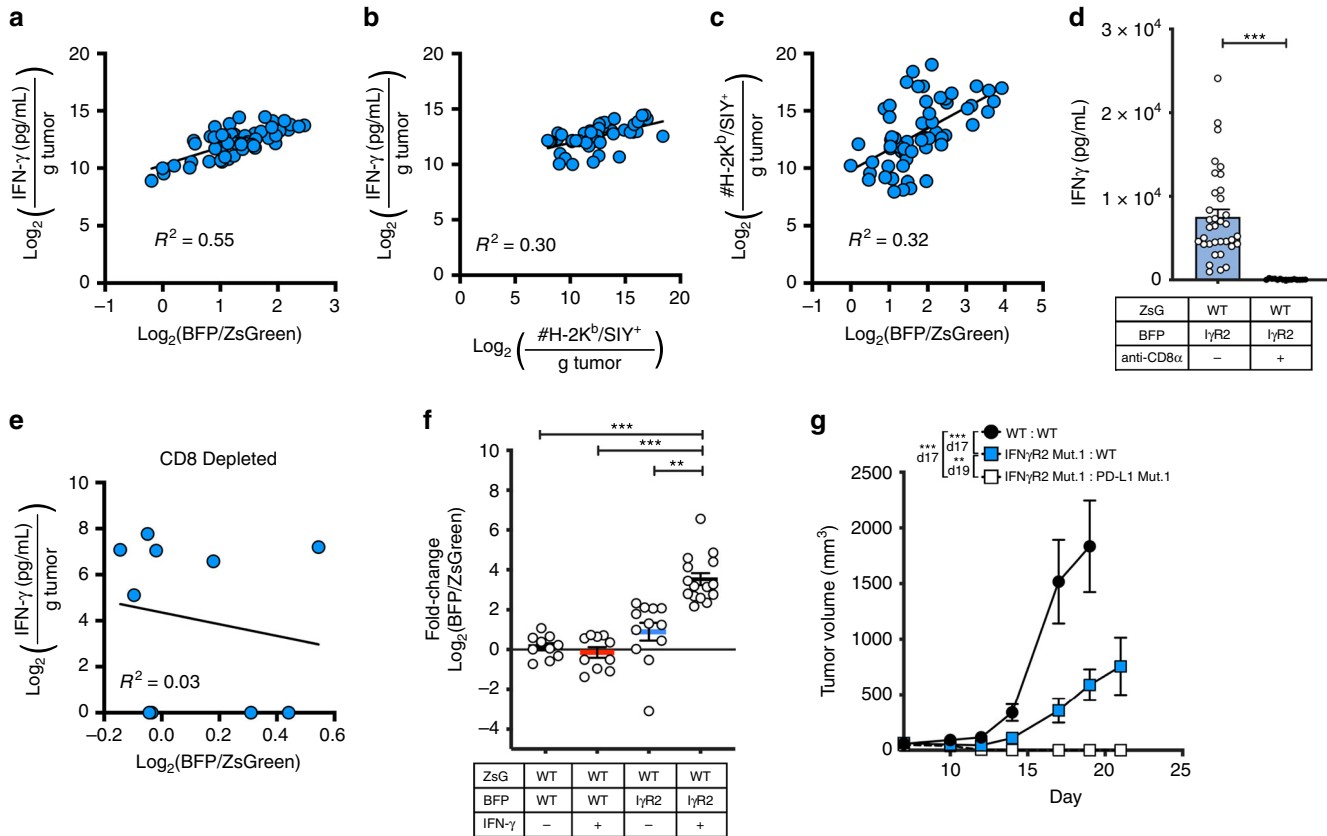

**Fig. 8 The antitumor effects of IFN-γ on WT tumor cells drives selection for IFNγR2-mutant tumor cells. a** and **b** Correlation between the concentration of intratumoral IFN-γ and the fold-change in the frequency of BFP$^+$ versus ZsGreen$^+$ tumor cells (**a**) and the total number of H-2K$^b$/SIY$^+$ CD8$^+$ TILs (**b**). **c** Correlation between the total number of H-2K$^b$/SIY$^+$ CD8$^+$ TILs and fold-change in the frequency of BFP$^+$ versus ZsGreen$^+$ tumor cells. **a–c** Tumors were analyzed on day 14 after tumor inoculation. All data were log2-transformed. $n = 60$ mice (**a, c**) and $n = 49$ mice (**b**); data are pooled from five independent experiments. **d** Concentration of IFN-γ after CD8$^+$ T cell depletion. $n = 35$ mice (no treatment) and $n = 15$ mice (anti-CD8α) **e** Correlation between intratumoral IFN-γ and log$_2$(BFP/ZsGreen) fold-change in the absence of CD8$^+$ T cells. Mice received 200 μg anti-CD8a (clone YTS169.4) on days -1 and 7. Tumors were analyzed on day 14. $n = 10$ mice; data are pooled from two independent experiments. **f** Fold-change in BFP$^+$ versus ZsGreen$^+$ tumor cells after administration of recombinant mouse IFN-γ. 50 μg of IFN-γ was administered i.p. on days 7, 10, 13, and 16. Tumor cell composition was analyzed on day 18. $n = 10$ mice (WT:WT and WT:WT + IFN-γ), $n = 12$ mice (WT:IγR2), and $n = 15$ mice (WT:IγR2 + IFN-γ); data are pooled from two independent experiments. **g** Tumor outgrowth of mixed WT and PD-L1-mutant tumors. $n = 10$ mice; data are pooled from two independent experiments. Results are expressed as mean ± s.e.m. Least squares regression was performed in (**a–c, e**). Statistical significance was determined by a Mann-Whitney test (**f**) and a two-way ANOVA Bonferroni post-hoc test (**g**). *$p < 0.05$, **$p < 0.01$, ***$p < 0.001$.

correlated with both selection of IFNγR2-mutant tumor cells and also the frequency of SIY/H-2K$^b$ CD8$^+$ TILs (Fig. 8a, b). Consistent with our previous observation, the total number of H-2K$^b$/SIY$^+$ CD8$^+$ TILs correlated with selection for IFNγR2-mutant tumor cells (Fig. 8c). Selection was mainly driven by IFN-γ produced by CD8$^+$ T cells as no selection was observed when CD8$^+$ T cells were depleted (Fig. 8d, e). To further test the role of IFN-γ in tumor cell selection, we reasoned that administration of additional IFN-γ could push the system further towards IFNγR2-muant tumor cells. To this end, we inoculated mice with WT:WT or WT:IFNγR2-mutant tumor cells and administered 50 μg of recombinant mouse IFN-γ i.p. on days 7, 10, 13, and 16 and analyzed tumor cell composition on day 18 after tumor cell inoculation. We observed a 3.9-fold increase in the frequency of IFNγR2-mutant tumor cells after IFN-γ administration compared to mice that did not receive IFN-γ (Fig. 8f).

Taken together, these results suggest a scenario in which IFN-γ-insensitive tumor cells gain a selective advantage by avoiding the direct antitumor effects of IFN-γ. This process likely occurs after the initial T cell insult diminishes to a point where PD-L1 expression on tumor cells is no longer required and can be compensated by PD-L1 on the mixed WT tumor

cells. To test whether PD-L1 on the admixed WT tumor cells was necessary for the selective outgrowth of IFNγR2-mutant tumor cells, the PD-L1 gene was disrupted using CRISPR. Indeed, under these conditions, progressive tumor growth was no longer observed, with no detectable outgrowth of IFNγR2-mutant tumor cells (Fig. 8g). These results indicate the involvement of clonal cooperation for immune selection of these mutants to occur, as WT tumor cells provide exogenous PD-L1 into the system. This impedes the immune response sufficiently to prevent total rejection of the mutant clones, allowing the differential effects of secreted IFN-γ to select for IFNγR2-mutants.

## Discussion
Our results utilizing an in vitro CRISPR screen revealed that tumor cell-intrinsic IFN-γ signaling is necessary for optimal T cell-mediated tumor cell killing in vitro. These results are consistent with other recently published CRISPR screens that similarly identified IFN-γ signaling-mutants using in vitro T cell selection systems[23,24]. However, resistance to T cell-mediated killing in vitro did not translate to aggressive tumor growth in vivo, as these IFN-γ-signaling-mutant tumors showed

improved immune-mediated tumor control (Supplementary Fig. 11). These discrepant phenomena may be explained by differing requirements for the levels of class I MHC upregulation required for tumor control in vitro versus in vivo, and also potentially distinct mechanisms of tumor cell destruction in the two settings. Short-term in vitro lysis assays are thought to be highly sensitive to the level of class I MHC expression by target cells, and cultured B16 melanoma cells show very low expression of H-2K$^b$ by flow cytometry in vitro. It is clear that IFN-γ produced by the CD8$^+$ effector T cells in vitro induces class I MHC upregulation that improves tumor cell lysis in a short-term assay. Following subcutaneous implantation into mice, tumor cells rapidly upregulate class I MHC, which is likely modulated by additional factors beyond IFN-γ. Mechanistically, in vitro lysis by CD8$^+$ T cells is dependent on perforin/granzyme release, yet we have previously published that T-cell-mediated control of B16 melanoma in vivo is intact in perforin-deficient mice[28]. Immunologic control of tumors in vivo can occur through additional killing mechanisms, such as engagement of death receptors[28–30], the antiproliferative effects of cytokines, both direct and through inhibition of angiogenesis[31,32], and even indirectly through the effects of macrophages or other innate immune cells[33,34]. Our use of CRISPR/Cas9 to delete H-2K$^b$ from tumor cells proved that, despite class I MHC expression being low on IFN-γ-signaling-mutant tumor cells, H-2K$^b$ was nonetheless required for anti-PD-L1 efficacy and CD8$^+$ T cell-mediated tumor control in vivo. Together, our results suggest that in vitro screens and in vivo confirmatory experiments offer complementary information that together can reveal the complexity of immune resistance within the tumor microenvironment.

Deeper analysis of the antitumor immune response to IFN-γ-insensitive tumors revealed an increased frequency of antigen-specific CD8$^+$ T cells within the tumor microenvironment of IFN-γ-insensitive tumors, while T cell priming systemically was only minimally affected. Three possibilities could explain the increase in H-2K$^b$/SIY$^+$ CD8$^+$ TILs: first, it could be that T cell recruitment to the tumor site is increased; second, intratumoral proliferative expansion of SIY-reactive CD8$^+$ TILs might be augmented; and third, CD8$^+$ TIL cell death might be diminished. Because the augmented antitumor immune effect required H-2K$^b$ expressed by tumor cells, it is likely that TIL proliferation and/or death are affected through this cognate interaction. IFN-γ has been shown to upregulate expression of FasL, which has been reported to contribute to TIL apoptosis in vivo[21]. In addition, IFN-γ upregulates expression of class II MHC, which in addition to presenting peptides to CD4$^+$ T cells also is a ligand for the inhibitory receptor LAG-3[35,36] expressed by tumor antigen-specific CD8$^+$ TILs[37]. Some of these considerations also may help to explain previous reports in which tumor cells expressing a dominate negative IFNγR1 exhibited increased tumor growth, as this dominate negative receptor can still bind IFN-γ and perhaps sequester it from other cells within the tumor. Future work will be required to continue to dissect the additional complexities of IFN-γ functions among the array of cells present within the tumor microenvironment.

The functional contribution of PD-L1 on tumor cells versus host APCs in engaging PD-1 on TILs and inhibiting T cell function has been controversial but likely varies with the tumor model or cancer histology being studied, and also the degree of antigenicity. In human cancer patients, T cell-infiltrated tumors can show PD-L1 upregulation on either tumor cells or myeloid cells, and each can be associated with anti-PD-1 therapeutic efficacy in defined contexts[38]. In mouse models, PD-L1$^{−/−}$ hosts have been reported to show improved antitumor immunity in some model systems[39,40], but in other models CRISPR/Cas9-mediated disruption of the PD-L1 gene in tumor cells can be

sufficient for improved tumor control[41]. In our current work, we utilize the B16.SIY and MC38 models, which have high antigenicity and prime strong CD8$^+$ T cell responses, even though the tumors eventually grow progressively. It now seems clear that a major mechanism for ultimate tumor growth in these models is through PD-L1 upregulation on tumor cells via persistent IFN-γ production in the tumor microenvironment. Our data highlight that tumor cells can afford to lose PD-L1 expression through disruption of the IFN-γ signaling pathway when PD-L1 expression on WT tumor cells can compensate for this loss.

The spontaneous in vivo control of IFN-γ-signaling-mutants in our current work seems initially to be at odds with other published reports showing a loss of IFN-γ signaling on tumor cells that grew out in association with acquired resistance to anti-PD-1 therapy[15,23–25]. While our final experiments studying selection following immunotherapy were performed with anti-PD-L1 and the available clinical data were obtained in the context of anti-PD-1 Ab therapy, both Abs are thought to function by a similar mechanism. We used single-cell cloning to establish our IFN-γ-insensitive tumor cell populations, therefore when engrafted into mice the starting tumor population lacked IFN-γ signaling on all tumor cells. In other experimental models, due to incomplete CRISPR/Cas9 mutagenesis, it is possible that a fraction of the starting tumor population retained IFN-γ signaling and was in fact a cellular mixture. Therefore, our in vivo results from the in vitro CRISPR screen might have been masked had we not taken the reductionist approach of deriving genetically homogenous tumor cell populations by single-cell cloning. One caveat of this method is that single-cell clones can remain subject to genetic drift as they expand in culture. However, reintroducing IFNγR2 in two tumor models restored progressive tumor growth indicating that the lack of IFN-γ signaling was the causative factor resulting in spontaneous tumor control.

The difference in growth rate between the B16F10 and B16.SIY models is likely a result of the differing immunogenicity between these tumors. Strong antigens that arise as a result of mutations can lead to T cell killing of immunogenic tumor cell populations[42]. Furthermore, the variable immunogenicity among clonal tumor cell populations is likely a determining factor in immunoselection, while the net immunogenicity of all tumor cells may dictate the relative contribution that immune-inhibitory pathways play in this process. Our collective data are consistent with the notion that strong immune selection for resistant mutants is dependent upon tumor cell antigenicity.

Clinically, tumors that grow out under immune selective pressure likely start out as a heterogeneous mixture, in which sensitive tumor cells are eliminated and resistant tumor cells grow progressively. To better align our mouse model with this clinical scenario, we found that when WT tumor cells were mixed with IFNγR2-mutant tumor cells and implanted in vivo, the IFNγR2-mutant cells were indeed selected out under strong immune pressure upon anti-PD-L1 therapy. Since PD-L1 is the major negative immune regulatory pathway upregulated by IFN-γ on tumor cells, once the PD-L1-/PD-1 interaction is masked, then it appears that the antitumor effects of IFN-γ dominate, selecting for IFN-γ-signaling-mutant tumor cells. Intratumor heterogeneity has several implications for the immunobiology of the tumor microenvironment. Perhaps most importantly, heterogeneity in the expression of mutant antigens (trunk versus branch mutations) can influence the degree to which a dominant T cell population may eliminate cancer cells following immunotherapy. Treatment with anti-PD-1 has been shown to prune the tumor cell population in clinical responders, as specific mutational epitopes become eliminated as assessed by on-treatment biopsies. Increased intratumor heterogeneity has been associated with inferior clinical activity of anti-PD-1 therapy in patients[43,44]. Our current work

suggests that intratumor heterogeneity in other molecular pathways also contributes to secondary resistance, beyond tumor antigen expression. In addition, our results suggest that two different tumor cell clones can cooperate to allow selective outgrowth of one clone under immune selective pressure.

## Methods

**Mouse cancer cell lines**. All cell lines were routinely tested for mycoplasma contamination using the HEKBlue (InvivoGen) reporter cell line, following the manufacturer's protocol. B16F10 (ATCC) cells were engineered to express DsRed fused in-frame with the model antigen SIYRYYGL[28,34]. The MC38 cell line was a gift from Dr. Yang-Xin Fu (UT Southwestern). DsRed expression was routinely monitored as it can shift over time, and flow cytometric sorting was performed periodically to ensure uniform expression.

To establish a founder B16.SIY.DsRed tumor cell line, single-cell B16.SIY.DsRed clones were tested for their relative intensity of DsRed expression compared to the B16.SIY.DsRed polyclonal population. Two B16.SIY.DsRed founder lines with comparable DsRed expression to the polyclonal B16.SIY.DsRed tumor cell population were tested for normal tumor growth and responsiveness to anti-PD-L1 + anti-CTLA-4 blockade. B16.SIY.DsRed clone #9 was chosen for subsequent experiments.

**Lentiviral packaging and transduction**. Individual sgRNA expression vectors were constructed as follows. Forward and reverse 26-nt oligonucleotides (Supplementary Table 2) were mixed at 10 mM each in 10 mM Tris-HCl (pH8.0) and 5 mM MgCl2 in a total volume of 100 µl. The mixture was incubated at 95 °C for 5 min and cooled to room temperature slowly to form duplex oligonucleotides. The duplex oligonucleotides were then cloned into the BbsI site of pKLV-U6sgRNA (BbsI)-PGKpuro2ABFP (Addgene, #50946). gRNA sequences targeting IFNγR2 and Jak1 were chosen from a genome-wide CRISPR library described in Koike-Yusa et al[26]. To generate the gRNA sequences for H-2K[b], the online ATUM-guide RNA design tool was used (www.atum.bio/eCommerce/cas9/input).

For Cas9 expression either a lentiviral vector encoding Cas9 and GFP (Sigma: pLV-U6g-EPCG) or a vector encoding Cas9, a piggyBac transposon element, and blasticidin resistance gene (Bsr) was used (Cas9-Bsr). Vector transfection was performed using Lipofectamine 3000 reagent (Thermo Fisher Scientific) according to the manufacturer's protocol. For lentivirus packaging, Two hundred and ninety-three T cells were cultured in complete DMEM without antibiotics at 50–70% confluence in a 6-well plate and were then transfected with lipofectamine particles containing the target vector, psPAX2 (Addgene, #12260) and pMD2.G (Addgene: #12259) packaging vector (Addgene, #12260). The day before viral supernatant collection, tumor cells were plated in 4 wells of a 24-well plate at 10,000 tumor cells per well. Forty-eight hours after transfection, supernatant was collected, filtered through a 0.45 µm filter, and added to tumor cells with polybrene at 10 µg/mL (1:1000 dilution). When cells became confluent they were transferred to a 100 × 15 cm[2] petri dish and expanded for three days. After expansion cells were sorted based on GFP (Cas9) or BFP (sgRNA) expression.

Genome-wide gRNA lentiviral library (Addgene, #50947) was packaged as described above in 293 T cells. Virus-containing media was collected 48 h post-transfection. $1 \times 10^5$ B16.SIY cells were seeded in each well of a 24-well plate one day before transduction. Five hundred microliter virus mixed with polybrene (8 µg/mL) was added to the B16.SIY cells. Transduction efficiency of B16.SIY cells was analyzed by flow cytometry based on BFP expression.

**Genome-wide CRISPR screen in B16.SIY tumor cells**. We generated a B16.SIY cell line that constitutively expressed Cas9 from a single-copy *piggyBac* transposon. Then the cell line was transduced with the genome-wide pooled gRNA library lentivirus, which contains 87,897 gRNAs targeting 19,150 mouse protein-coding genes. To adequately represent all gRNAs in the library, each independent screen we infected $2 \times 10^7$ cells with a transduction efficiency over 90%. After transduction, B16.SIY cells were selected with puromycin for 3 days followed by a 5 day expansion. In a single screen, $5 \times 10^7$ B16.SIY cells were co-cultured with pre-activated 2 C/Rag2$^{-/-}$ T cells at 1:2 ratio overnight (16 h). 2 C T cells were stimulated with plate-bound anti-CD3ε (1 µg/mL) and anti-CD28 (2 µg/mL) in non-tissue culture treated 6-well plates and supplemented with 100 U of IL-2. The T cells were removed and the remaining B16.SIY cells were allowed to recover for 1 week. Two independent screens were performed. Genomic DNA was extracted from live cells by phenol/chloroform purification. One microgram genomic DNA was used in five independent PCR reactions using the primers 5'-agtttggttag-taccgggcc-3' (gRNA F1) and 5'-gatccaaaaaaagcaccgac-3' (gRNA F2). The PCR products were pooled and 1 µg was cut with ApaI and BamHI and ligated into the pKLV-U6sgRNA(BbsI)-PGKpuro2ABFP by Gibson assembly (NEB). The ligation mixture was transformed into Stbl3 competent cells and 120 individual colonies from LB agar plates (50 µg/mL Ampicillin) were sequenced using the primer 5'-gagggcctatttcccatgatt-3' (hU6F).

To validate IFNγR2 and Jak1, IFNγR2- and Jak1-mutant B16.SIY tumor cells, cells were plated at a density of 10,000 cells per well in a 24-well plate and allowed to adhere overnight. Pre-stimulated 2 C T cells were titrated and transferred into

wells. After 24 h, killing was measured by either visibly counting remaining adhered tumor cells or by trypsonizing tumor cells and staining for dead cells with a live/dead discrimination dye. Live cells were considered to be dsRed$^+$ and live/dead$^-$ and were enumerated using CountBright beads (FISHER).

**Single-cell cloning**. Single-cell sorting of tumor cells was performed using a FACS Aria II cell Sorter (BD Biosciences). Cells were sorted into single wells of a 96-well plate containing complete DMEM media (10% FBS, 0.01 M MOPS, 100 U/mL penicillin/streptomycin, 1% MEM Non-Essential Amino Acids). Visible colonies from single cells occurred after ~2 weeks. After single-cell clones reached confluence they were transferred to a 24-well plate and 6–18 clones from each single-cell sort were further examined. After 1–2 days of expansion, an aliquot of each clone was tested for disruption of IFNγR2 or Jak1 through the failure to upregulate H-2K[b] and/or PD-L1 in response to IFN-γ stimulation. Responsiveness to IFN-γ was performed in a 96-well plate and stimulated with IFN-γ (5 ng/mL) for 16–20 h followed by flow cytometric evaluation. Single-cell clones that exhibited disruption of the target gene were then expanded in a $100 \times 15 \text{ cm}^2$ petri dish followed by an expansion in a 225 cm[2] flask. Stocks of cell lines were frozen and stored at $-150$ °C.

**Generation of targeted deletion mutants**. To generate loss-of-function mutations in the IFNγR2 and Jak1 genes, the B16.SIY.DsRed founder line was transduced with lentivirus encoding the targeting gRNA and a BFP reporter. Cells were allowed to recover for 3 days and then transfected with the Cas9-Bsr plasmid. Cells were selected with blasticidin for 2 days and then single-cell sorted based on BFP expression. To confirm successful disruption of Ifngr2 and Jak1 the targeted region was amplified by PCR, TA-cloned (Invitrogen, TOPO-TA), and transformed into Stbl3 chemically competent *E. Coli*. Sequencing was performed using M13R primer on 10–15 bacterial colonies to obtain sequences for both alleles. The primers used for amplification of the targeted region in Ifngr2 and Jak1 are shown in Supplementary Table 2.

To generate IFNγR2/H-2K[b] double-mutant cells, #9 B16.SIY cell line was transfected with both Cas9-GFP (Sigma) and pKLV-U6-sgRNA(H2kb)-PGKpuro2ABFP vectors. Cells were single-cell sorted for BFP and GFP expression. Clones were then tested for the inability to upregulate H-2K[b] in response to IFN-γ (5 ng/mL) stimulation. One clone was chosen, expanded and transfected with Cas9-GFP and pKLV-U6-sgRNA(Ifngr2)-PGKpuro2ABFP vectors. Cells were sorted and tested for the inability to upregulate PD-L1 in response to IFN-γ stimulation. The #9 B16.SIY IFNγR2 H-2K[b] double-mutant cell line was then transduced with the empty vector pKLV-U6-PGKpuro2ABFP lentivirus to equalize BFP expression to the IFNγR2 Mut.1 tumor cells.

For experiments using MC38 and B16F10 tumor cells, Cas9-GFP (Sigma: pLV-U6g-EPCG) was retrovirally introduced and cell sorted 2-3 times for GFP expression. To generate loss-of-function mutations in the IFNγR2 and Jak1 genes, cells were transfected with the corresponding pKLV-U6-sgRNA-PGKpuro2ABFP vectors and single-cell sorted based on GFP and BFP expression and tested for disruption of the targeted gene by IFN-γ (5 ng/mL) stimulation.

To reintroduce IFNγR2 into IFNγR2-mutant cell lines, a codon-optimized IFNγR2 gBlock (IDT) with mutated gRNA sites and flanking BamHI and NotI cut sites was digested and inserted into the pRetro-IRES-ZsGreen1 (Clontech) expression vector. For Jak1, due to the large size of the Jak1 gene, three codon-optimized gBlock fragments were synthesized and assembled into the pRetro-IRES-ZsGreen1 expression vector by Gibson assembly. Sequences for gBlocks are in Supplementary Fig. 10. Reintroduction of Jak1 into the Jak1 Mut.1 tumor cell line did not restore IFN-γ signaling, possibly due to a dominant negative effect from the endogenous mutated Jak1 protein. However, reintroduction of Jak1 into the Jak1 Mut.2 cell line did restore IFN-γ signaling and was used for subsequent experiments.

**Mice and tumor inoculation**. C57BL/6NTac mice were purchased from Taconic farms. Rag2$^{-/-}$ (NCI) mice were bred in house under specific pathogen-free conditions. 2 C/Rag2$^{-/-}$ mice were bred in our facility. For tumor inoculation, unless otherwise stated, $2 \times 10^6$ tumor cells were subcutaneously injected into the right flank of female mice. Tumor volume was measured around twice per week using calipers. Tumor volume was calculated: $T_V = T_L \times T_W \times T_H$, where $T_L$ is the tumor length, $T_H$ is tumor height, and $T_W$ is tumor width. All mice were maintained according to the National Institute of Health Animal Care guidelines and utilized under IACUC-approved protocols. All experiments were approved by the Institutional Animal Care and Use Committee at the University of Chicago and followed international guidelines.

For IFNγR2 Mut.1 and WT mixing experiments, $1 \times 10^6$ of each tumor cell line was used for a total of $2 \times 10^6$ tumor cells. The ratio of tumor cells was analyzed by flow cytometry before tumor inoculation to ensure a near 50:50 percent ratio. Mixed tumor cells were inoculated into Rag2$^{-/-}$ mice to control for differences in tumor cell growth rate and viability. The BFP/ZsGreen ratio from Rag2$^{-/-}$ mice was used to normalize the ratio from experimental groups.

**Antibody administration**. For antibody treatments, mice received 100 µg of anti-PD-L1 (10 F.9G2: BioXCell) and/or 100 µg anti-CTLA-4 (BE0131: BioXCell) intraperitoneally on days 7, 10, and 13 after tumor engraftment. For depletion

studies, mice received 200 µg of either anti-CD8 (YTS 169.4: BioXCell) and/or anti-NK1.1 (PK136: BioXCell) or a combination of anti-CD8+ anti-NK1.1 2 days before tumor engraftment and once every 7 days after. Depletion of the relevant cell populations was confirmed by flow cytometry analysis of blood 3 days after antibody administration and in the tumor at endpoint.

**TIL and tumor cell isolation.** Tumors were harvested from mice at the indicated time points. Tumors were digested using an enzyme mix containing Collagenase (1 mg/mL, Sigma: C5138), DNAse (200 µg/mL, Sigma: D5025), and Hyaluronidase (100 µg/mL, Sigma: H6254) for 20–30 min at 37ºC while rotating. The tumor suspension was filtered through a 50 µm filter and washed with PBS and purified by Ficoll-Hypaque density gradient centrifugation. For TIL sorting, cells were washed twice with PBS before staining and maintained in complete DMEM (cDMEM) (cDMEM: 10% FBS, 100 U/mL Penicillin-Streptomycin, 1% MEM Non-Essential Amino Acids, 50 µM β-mercaptoethanol, 0.01 M MOPS) and sorted into RLT buffer using an Aria II flow cytometric cell sorter (BD). Sorted cells were rapidly frozen on dry ice. Gating strategy for cell sorting experiments are represented in Supplementary Fig. 8.

**Flow cytometry and antibodies.** For samples not sorted, cells were washed with PBS and stained in FACS buffer (2% FBS, 2 mM EDTA, and 0.001% NaN$_3$). All gating strategies are represented in Supplementary Fig. 9. Cells were first stained for H-2K$^b$/SIY-pentamer (PE; ProImmune) for 10 min at room temperature at a 1:20 dilution, followed by staining with remaining antibodies for 20 min on ice. Antibodies against the following molecules were used: CD3ε (clone 17A2, BioLegend, 100216), Thy1.2 (clone 30-H12, BioLegend, 105320), CD45.2 (clone 104, BioLegend, 109806), CD8α (clone 53-6.7, BioLegend, 100747), CD4 (clone RM4-5, BioLegend, 100547), PD-L1 (clone 10 F.9G2, BioLegend, 124312), CD19 (clone 6D5, BioLegend, 115545), I-A/I-E (clone M5/114.15.2, BioLegend, 107630), and H-2K$^b$ (clone AF6-88.5, BioLegend, 742862). All antibodies were used at a 1:100 dilution. Fixable Viability Dye eFluor 506 or 780 (eBioscience) was used for live/dead discrimination and was used at a 1:200 dilution. All flow cytometric analysis was conducted on either an LSRFortessa or X20 (BD) and analyzed using FlowJo software (Tree Star).

**Quantitative real-time PCR and RNA sequencing.** Total RNA was extracted from sorted cell populations using the RNeasy Micro Kit (QIGEN) following manufacturers protocol. For qRT-PCR, cDNA was synthesized using the High Capacity cDNA Reverse Transcription kit (Applied Biosystems) according to manufacturer's protocol. Transcript levels were measured using primer-probe sets (Supplementary Table 2) developed through the online ProbeFinder Software (Roche).

RNA sequencing was performed by the University of Chicago Genomics Core facility using the Illumina HiSeq platform. Alignment was performed using the Kallisto software. All submitted samples passed quality control using the R package FastQC. Raw read counts were processed by TMM normalization followed by log$_2$ transformation. Genes expressed in fewer than three samples were removed from further analysis. Limma voom was used to identify differentially expressed genes between each mutant cohort and the WT cohort with a cutoff fold-change > 2 and adjusted p-value < 0.05. RNASeq data are available in the Gene Expression Omnibus database under accession no. GSE125698.

**IFN-γ ELISA.** For IFN-γ ELISA, tumors were isolated, separated into two sections and weighed. One section was dissociated using frosted slides in 500 µl cold PBS and pelleted at 21,130 × g for 10 min at 4 °C. Fifty microliter of supernatant was used for IFN-γ ELISA (R&D) following manufacturer's protocol.

**Reporting summary.** Further information on research design is available in the Nature Research Reporting Summary linked to this article.

## Data availability

The datasets generated during and/or analyzed during the current study are available in the Article, Supplementary Information files or available from the corresponding author on reasonable request. RNASeq data can be found in the NCBI Gene Expression Omnibus under access code GSE125698. The source data underlying graphs in Figs. 1–8 and Supplementary Figs. 2-7 has been provided as a source data file.

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

## Acknowledgements

This work was supported by R35CA210098 and a CBC Catalyst Award. J.B.W. was funded by Immunology Training Grant T32 AI07090. The flow cytometry core facility and genomics core facility at the University of Chicago are supported by a Cancer Center Support Grant (P30CA014599). We would like to acknowledge Drs. Maria-Luisa Alegre and Justin Kline for insightful comments about this study, and Yesika Contreras Duarte for her assistance with sample processing and data acquisition.

## Author contributions

S.L, H.H. and T.F.G. conceived of the project with assistance from J.B.W; J.B.W., S.L., H.H. and T.F.G. designed the project and wrote the manuscript with assistance from A.C. and E.F.H; J.B.W., S.L. and T.F.G designed experiments; J.B.W. and S.L. performed experiments and acquired data with assistance from A.C. and E.F.H; E.F.H. performed the RNASeq analysis and visualization. X.W. provided the Cas9 plasmid and helped design the initial CRISPR screen. T.F.G. and H.H. contributed equally.

## Competing interests

The authors declare no competing interests.
