## [Peer Review File · Nature Communications]

Reviewers' comments:

Reviewer #1 (Remarks to the Author):

This is an interesting set of studies that support the role of IFN-g signaling as a mechanism to allow a cancer cell-reactive immune resistance, likely through the expression of PD-L1 and maybe also IDO. The studies are conducted with three different tumor models and with more than one crispr knock-down clone for some of the studies.

Major comments:

The title of the article, the general focus and the conclusions are misleading and do not reflect the data that is being presented. Citing "immune resistance" in the title implies indirectly that the model includes an active form of immunotherapy, but it does not as the presented data refers to the rejection of subcutaneously implanted tumor cells. This is different from becoming resistant to anti-PD-1/L1 therapy, which is the underlying misleading focus of the article.

There are multiple directly misleading comments that try to frame the work in the wrong context. For example "However, these results are at odds with recently published cases of acquired immune resistance in patients with metastatic melanoma treated with anti-PD-1." Why are these results at odds with patients with acquired resistance (relapse after a period of response) to anti-PD-1? They are not, as this article does not seem to have a single experiment with PD-1 blockade therapy with the IFNgR signaling KO cell lines, so it does not test either primary or acquired resistance to PD-1 blockade therapy. All of these comments should be deleted and the focus of the article and its conclusions should stick to the data presented.

The unifying interpretation of the presented data is that IFNgR is important for subcutaneously implanted immunogenic cancer cells to escape from the initial T cell response. This is consistent with the role of IFN-g-inducible PD-L1 expression in providing protection of cancer cells from antitumor T cells. It is clear that this happens in human cancers as before the advent of PD-1 blockade therapy, these cancers would grow progressively with PD-L1 expression keeping antitumor T cells in an exhausted phenotype without attacking the cancer. Once the PD-1:PD-L1 interaction is blocked by an anti-PD-1/L1 antibody therapy, then the cancers that were being protected by reactive PD-L1 expression respond to therapy. In a way, the authors have re-discovered that IFN-gamma signaling results in reactive PD-L1 expression and protection from T cells, which has been termed adaptive immune resistance. Indeed, the author's data would be of interest if presented this way as it provides a nice model to demonstrate its role in vivo. Furthermore, one of the author's conclusions that this work supports the importance of cancer-intrinsic PD-L1 expression, as opposed to PD-L1 expression by other host cells in the tumor microenvironment, could be further developed as it is a solid and important conclusion from this work.

If the authors want to make the claim that their work suggests that lack of IFNgR signaling is beneficial to cancer cells in the setting of PD-1 blockade therapy, then they could take the B16-IFNgR2 Mut1 or the B16-JAK1 Mut1 from Supplemental Figure 3d, which grow progressively in mice, and give anti-PD-1 or anti-PD-L1 therapy after they are established and are palpable. If in this setting they find that these two tumors significantly respond to the therapy, they it would make sense to keep their title and comments about the implications of this work on primary or acquired resistance to PD-1 blockade therapy. However, it is unlikely that they will find responses in these with anti-PD-1/L1 in these tumors.

Another way to test if the postulate of the authors is correct would be to give a much higher inoculum of B16-SIY or MC38 tumor cells with the IFNgR2 or JAK1 KO and see if the tumors grow progressively. The prediction would be that at some point, the protective benefit of IFNgR signaling to reactively express PD-L1 (and/or IDO) would be lost, and tumors would grow

progressively. Again, if in this setting they could show anti-PD-1/L1 responses, then this would have implications on primary or acquired resistance to PD-1 blockade therapy.

Minor comments:

The authors could consider including additional references on the role of JAK1/2 mutations in primary and acquired resistance to PD-1 blockade therapy and not repeatedly state that it is based on two cases from the Zaretsky et al. NEJM 2016 article. These would be Sucker et al Nature Communications 2017, and Shin et al. Cancer Discovery 2018.

In addition, for completeness and considering that they are submitting to Nature Communications, they could also reference the Sade-Feldman et al. Nature Communications 2017 article that reports on several cases of B2M loss in resistance to PD-1 blockade therapy.

Reviewer #2 (Remarks to the Author):

Summary:

The authors carry out a genome-wide CRISPR screen in B16.SIY melanoma cells and confirm *Ifngr2* and *Jak1* as important genes conferring sensitivity to T cell mediated killing in vitro. However, when implanted into mice, the *Ifngr2* and *Jak1* mutant tumors were sensitized to anti-tumor efficacy of CD8+ T cells, which was mapped to defective Pd-I1/Cd274 up-regulation on mutant tumor cells. The authors then did an interesting experiment where they mixed wild-type and mutant (either *Ifngr2* or *Jak1*) tumor cells at 1:1 ratio, implanted these into mice and found that the mutant tumors would grow out, demonstrating the complexity of functions for IFN γ in anti-tumor immunity and showing that intra-tumor heterogeneity can contribute to patterns of immunotherapy.

Comment:

The paper is well written and presented clearly. Methods are rigorous. The screen didn't really appear to work very well, given published results from other groups that identify complete IFGR/JAK/STAT pathway genes [e.g. Manguso et al, 2017, Nature; Pan et al, 2018, Science; Kearney et al, 2018, Sci Immunol].

The major criticism of this manuscript is that the take home message from the paper needs revision. The fact is that the in vitro pooled screens ended up precisely predicting the final in vivo results, as well as what is seen clinically. This message needs significant improvement and the authors should consider changing the title. In the discussion of the manuscript, the authors are critical of the in vitro screens, when in fact the in vivo model systems using pure knockout backgrounds led to technical artifacts that don't translate. The pooled in vitro screens predicted perfectly what was seen in vivo when the appropriate experiment (i.e. mixed populations, strong CD8 challenge) was performed. The message should be that carefully done in vitro screens translate well when considering how they were performed. Injecting homogenous knockout populations to assess resistance is NOT the correct approach in mimicking a heterogenous mixed population of cells in vitro. It is also not representative of tumors. Thus, it is not a surprise that the mixing of wild-type and knockout tumor cells followed by injection re-capitulates the results of the pooled screen. Despite my lack of enthusiasm for some of the messaging, this study presents an important methodological advance and important contribution/warning for the field that warrants publication.

The argument to use knockout clones is acceptable. However, clonal knockout lines are still subject to both genetic and more importantly phenotypic drift by transcriptional and proteome changes [see Liu et al, 2019, Nature] and these challenges should be addressed more thoroughly in the discussion.

Additional mechanistic studies would go a long way to strengthening the observation that Ifngr2 homogenous mutant tumors are well selected by CD8 cells. For example, what drives these cells immunogenicity at baseline? Is it possible that this is an artifact of transplantable mouse models that develop an inflammatory reaction after subcutaneous injection? It is not at all clear that this data would translate to human cancers. However, the technical and methodological contributions of this data are still solid.

Minor points:

Enriched hits from the screen could be presented in a more quantitative fashion.

Supp Figure 3 and 5 legends need to be switched.

Methods for the validation of T-cell killing assays could be better detailed.

The methods for tumor digestion and flow cytometry experiments outline in Fig 7 need to be included and detailed.

Dear editors:

We appreciate the thoughtful review of our manuscript, and are pleased that there is support for publication. We have addressed the reviewers' comments with additional experiments and text edits, as outlined in our in a point-by-point response below. Because of new data and reviewer suggestions, we have importantly changed the title of the article to "Immune selection for IFN- γ signaling mutant cancer cells involves tumor heterogeneity and clonal cooperation".

Reviewer #1 (Remarks to the Author):

Major comments:

Comment 1:

The title of the article, the general focus and the conclusions are misleading and do not reflect the data that is being presented. Citing "immune resistance" in the title implies indirectly that the model includes an active form of immunotherapy, but it does not as the presented data refers to the rejection of subcutaneously implanted tumor cells. This is different from becoming resistant to anti-PD-1/L1 therapy, which is the underlying misleading focus of the article.

We changed the title to better reflect the findings in the manuscript, incorporating new data as described below. The new title is "Immune selection for IFN- γ signaling mutant cancer cells involves tumor heterogeneity and clonal cooperation"

There are multiple directly misleading comments that try to frame the work in the wrong context. For example "However, these results are at odds with recently published cases of acquired immune resistance in patients with metastatic melanoma treated with anti-PD-1." Why are these results at odds with patients with acquired resistance (relapse after a period of response) to anti-PD-1? They are not, as this article does not seem to have a single experiment with PD-1 blockade therapy with the IFN γ R signaling KO cell lines, so it does not test either primary or acquired resistance to PD-1 blockade therapy. All of these comments should be deleted and the focus of the article and its conclusions should stick to the data presented.

We thank the reviewer for this raising this potentially confusing aspect in our manuscript. We have amended the flow of text in the argument, but the fact of the matter is that the fundamental result here is indeed at odds with the published reports cited. In our study tumor cells with defective tumor cell-intrinsic IFN γ R signaling were better controlled immunologically, not less-well controlled. In our revised paper, we have continued the effort at reconciliation by extending mixing experiments along with PD-L1/PD-1 blockade. These new data that are now included generate the interesting idea that not only tumor heterogeneity but also clonal cooperation are necessary to select for IFN γ R signaling mutants upon immunotherapy treatment in vivo. The wildtype tumor cells bring in PD-L1, which slows down the immune response and enables immune selection to occur. We also have shown that the mechanism involves IFN- γ production by anti-tumor T cells. These results also helped guide the re-phrasing of the title. The new concepts are represented in the graphical abstract/cartoon included in this document.

The unifying interpretation of the presented data is that IFN γ R is important for subcutaneously implanted immunogenic cancer cells to escape from the initial T cell response. This is consistent with the role of IFN-g-inducible PD-L1 expression in providing protection of cancer cells from antitumor T cells. It is clear that this happens in human cancers as before the advent of PD-1 blockade therapy, these cancers would grow progressively with PD-L1 expression keeping antitumor T cells in an exhausted phenotype without attacking the cancer. Once the PD-1:PD-L1 interaction is blocked by an anti-PD-1/L1 antibody therapy, then the cancers that were being protected by reactive PD-L1 expression respond to therapy. In a way, the authors have re-discovered that IFN-gamma signaling results in reactive PD-L1 expression and protection from T cells, which has been termed adaptive immune resistance. Indeed, the author's data would be of interest if presented this way as it provides a nice model to demonstrate its role in vivo.

We take note of the reviewer's point that upregulation of PD-L1 is an important known adaptive resistance mechanism and made this clearer in the text. Changes can be found on page 3, 8, and 9. We believe that along with our new experimental data mentioned above, that our overall revised interpretation is consistent with the reviewer's comments.

Furthermore, one of the author's conclusions that this work supports the importance of cancer-intrinsic PD-L1 expression, as opposed to PD-L1 expression by other host cells in the tumor microenvironment, could be further developed as it is a solid and important conclusion from this work.

We agree that our data indicates that PD-L1 expression on tumor cells is critical for ultimate tumor growth, especially as an adaptive immune resistance mechanism to block the initial T cell insult, at least in these mouse models. In addition, our data also indicate the tumor cells can afford to lose PD-L1 as shown in our tumor mixing experiments. It is likely that loss of PD-L1 expression can only occur when compensatory mechanisms are in place, for instance, when PD-L1 is expressed on other wildtype tumor cells. This discussion point was added and can be found on pages 13 and 14.

If the authors want to make the claim that their work suggests that lack of IFN γ R signaling is beneficial to cancer cells in the setting of PD-1 blockade therapy, then they could take the B16-IFN γ R2 Mut1 or the B16-JAK1 Mut1 from Supplemental Figure 3d, which grow progressively in mice, and give anti-PD-1 or anti-PD-L1 therapy after they are established and are palpable. **If in this setting they find that these two tumors significantly respond to the therapy, then it would make sense to keep their title and comments about the implications of this work on primary or acquired resistance to PD-1 blockade therapy.** However, it is unlikely that they will find responses in these with anti-PD-1/L1 in these tumors.

We believe that our mixing experiment to recapitulate human tumor heterogeneity clarifies these issues. When clonal, IFN γ R signaling mutants are better controlled because there is no tumor cell-derived PD-L1 in the system. But when admixed with WT tumor cells, anti-PD-L1 now selects for outgrowth of the mutants. We have also shown that the WT tumor cells no longer promote outgrowth of the mutant tumor cells if they lack PD-L1, demonstrating that exogenous PD-L1 is a major factor brought in by the admixed WT tumor cells. We also have added mechanistic data around T cell-derived IFN- γ being a major factor in this selection process.

Another way to test if the postulate of the authors is correct would be to give a much higher inoculum of B16-SIY or MC38 tumor cells with the IFN γ R2 or JAK1 KO and see if the tumors grow progressively. The prediction would be that at some point, the protective benefit of IFN γ R signaling to reactively express PD-L1 (and/or IDO) would be lost, and tumors would grow progressively. Again, if in this setting they could show anti-PD-1/L1 responses, then this would have implications on primary or acquired resistance to PD-1 blockade therapy.

Although we have inoculated over 50 mice with IFN γ R2-mutant B16.SIY tumors and a few have escaped the immune response, all of these were antigen-loss variants, which was measured by loss of dsRed expression. So this experimental setup is not ideal for our intended purpose. Rather, our approach to mimic intratumor heterogeneity, as is likely occurring in patients, does indeed select for the mutant cells, consistent with clinical observations. And our new data mentioned above tie in the mechanism of this effect.

Minor comments:

The authors could consider including additional references on the role of JAK1/2 mutations in primary and acquired resistance to PD-1 blockade therapy and not repeatedly state that it is based on two cases from the Zaretsky et al. NEJM 2016 article. These would be Sucker et al Nature Communications 2017, and Shin et al. Cancer Discovery 2018.

Suggested references have been added where acquired resistance to PD-1 blockade therapy is mentioned. See pages 3, 5, 11, and 14.

In addition, for completeness and considering that they are submitting to Nature Communications, they could also reference the Sade-Feldman et al. Nature Communications 2017 article that reports on several cases of B2M loss in resistance to PD-1 blockade therapy.

Reference has been added. See page 3.

Reviewer #2 (Remarks to the Author):

Summary:

The authors carry out a genome-wide CRISPR screen in B16.SIY melanoma cells and confirm *Ifngr2* and *Jak1* as important genes conferring sensitivity to T cell mediated killing in vitro. However, when implanted into mice, the *Ifngr2* and *Jak1* mutant tumors were sensitized to anti-tumor efficacy of CD8+ T cells, which was mapped to defective Pd-11/Cd274 up-regulation on mutant tumor cells. The authors then did an interesting experiment where they mixed wild-type and mutant (either *Ifngr2* or *Jak1*) tumor cells at 1:1 ratio, implanted these into mice and found that the mutant tumors would grow out, demonstrating the complexity of functions for IFN γ in anti-tumor immunity and showing that intra-tumor heterogeneity can contribute to patterns of immunotherapy.

Comment:

The paper is well written and presented clearly. Methods are rigorous. The screen didn't really appear to work very well, given published results from other groups that identify complete IFGR/JAK/STAT pathway genes [e.g. Manguso et al, 2017, Nature; Pan et al, 2018, Science; Kearney et al, 2018, Sci Immunol].

In fact, our screen has additionally revealed some novel molecular aberrations, which are being pursued mechanistically in ongoing work. We chose to focus the current report on the unexpected findings around IFN γ R signaling mutants.

The major criticism of this manuscript is that the take home message from the paper needs revision. The fact is that the in vitro pooled screens ended up precisely predicting the final in vivo results, as well as what is seen clinically. This message needs significant improvement and the authors should consider changing the title. In the discussion of the manuscript, the authors are critical of the in vitro screens, when in fact the in vivo model systems using pure knockout backgrounds led to technical artifacts that don't translate. The pooled in vitro screens predicted perfectly what was seen in vivo when the appropriate experiment (i.e. mixed populations, strong CD8 challenge) was performed. The message should be that carefully done in vitro screens translate well when considering how they were performed. Injecting homogenous knockout populations to assess resistance is NOT the correct approach in mimicking a heterogenous mixed population of cells in vitro. It is also not representative of tumors. Thus, it is not a surprise that the mixing of wild-type and knockout tumor cells followed by injection re-capitulates the results of the pooled screen. Despite my lack of enthusiasm for some of the messaging, this study presents an important methodological advance and important contribution/warning for the field that warrants publication.

We are pleased the reviewer thinks our manuscript warrants publication. We did not intend to diminish the research capabilities of in vitro CRISPR screens or raise any uncertainty regarding results obtained from these screens. We only want to encourage conformational studies and deeper investigations of in vitro CRISPR screen results using in vivo models. We have changed the language of the messaging to not lessen the functionality of in vitro CRISPR screen. See pages 13 and 15. Further, the reviewer is correct in saying that had we chosen a polyclonal population of CRISPR targeted tumor cells we might better recapitulate the clinical scenario. However, if we had not taken the more reductionist approach of single cell cloning we would have missed the complex role IFN- γ signaling plays during different phases of the antitumor immune response, in particular the increased immune control based on lack of PD-L1 upregulation. And our new observation that clonal cooperation is involved, as the wildtype cells re-introduce PD-L1 into the equation, we believe provide a solid mechanistic explanation for the phenomenon. Deliberate reconstitution of heterogeneity, we believe, addresses this reviewer concern.

The argument to use knockout clones is acceptable. However, clonal knockout lines are still subject to both genetic and more importantly phenotypic drift by transcriptional and proteome changes [see Liu et al, 2019,

Nature] and these challenges should be addressed more thoroughly in the discussion.

Of course we agree that all tumor cell lines continue to have phenotypic drift over time. For this reason, we regularly sort our transduced cells for uniform expression of GFP, DsRed, etc. This part of our approach has been added to the Methods. The possibility that additional variation and downstream transcriptional changes secondary to CRISPR deletion need to be taken into account in these types of experiments has been added to the Discussion.

Additional mechanistic studies would go a long way to strengthening the observation that *lfngr2* homogenous mutant tumors are well selected by CD8 cells. For example, what drives these cells immunogenicity at baseline? Is it possible that this is an artifact of transplantable mouse models that develop an inflammatory reaction after subcutaneous injection? It is not at all clear that this data would translate to human cancers. However, the technical and methodological contributions of this data are still solid.

We have expanded the manuscript to provide more insight into the role $IFN-\gamma$ signaling plays in adaptive resistance and immune-mediated selection. $IFN-\gamma$ protein in the tumor microenvironment correlates with immune selection. Our new data also support a mechanism involving interclonal cooperation in the selection of $IFN-\gamma$ -insensitive tumor cells. Briefly, we show that $IFN-\gamma$ -insensitive tumor cells escape the anti-tumor effects of $IFN-\gamma$, but this can only occur if WT cells are present to provide PD-L1 to slow down T cell-mediated tumor destruction. We believe that our new results may provide an understanding as to why some human tumors select for $IFN-\gamma$ -signaling mutant tumor cells despite failed PD-L1 upregulation in response to $IFN-\gamma$. These data also address the reviewer's concern regarding cloned populations on CRISPR-edited cells.

Minor points:

Enriched hits from the screen could be presented in a more quantitative fashion.

Supp Figure 3 and 5 legends need to be switched.

Methods for the validation of T-cell killing assays could be better detailed.

The methods for tumor digestion and flow cytometry experiments outline in Fig 7 need to be included and detailed.

Supplemental Figure 1 has been updated to indicate the frequency of gRNA recovery.

Figure legends for supplemental 3 and 5 have been switched. We thank the reviewer for catching this error.

Method detailing 2C-mediated T cell killing to validate genetic targets has been added.

Additional changes:

In addition to the edits derived from reviewer suggestions, we have changed the title and overall message of the manuscript. We also added an experiment in which $IFN\gamma R2$ was re-introduced into $IFN\gamma R2$ KO MC38 cells, to confirm reversion of the phenotype.

Reviewers' comments:

Reviewer #1 (Remarks to the Author):

The authors provide an improved resubmission addressing several of the comments from the initial review. There are additional issues to address:

Major comments:

The introduction still refers to "two melanoma patients" regarding deficiency of IFN γ R signaling pathway and resistance to PD-1 blockade therapy, despite the authors having added other relevant references with additional cases. The text should be equally corrected to match the new references.

The B16.SIY is a very immunogenic variant of B16, with the inclusion of the SIY and DsRed foreign proteins. This is a model skewed towards a particular strong immune phenotype and the conclusions should reflect this point. The interpretation of the results is improved with the resubmission, but some of the conclusions continue to be too broadly stated to correctly reflect the data.

The studies using B16 in supplemental figure 3 are performed in a setting of a suboptimal number of implanted cells to maintain the author's argument that their data with B16.SIW.DsRed and MC38 is replicated in a lower immunogenicity model. However, this is not the case. This experiment should be performed with a regular B16 cell inoculum and added to the main figures as opposed to being relegated to the supplemental materials. The text should reflect that it barely replicates the data with the highly immunogenic models used for the other studies.

The text in page 9 continues to make the case that the author's data is at odds with findings in patients, but it is not as patients receive anti-PD-1 therapy and there is no such experiment in this work. Same comment is relevant for line 350.

The comment in lines 372-374 about the need to replicate data from in vitro screens with in vivo screens is fair, but skewed by the author's experience in interpreting their in vitro screen results in a way that did not anticipate a different result when tested in vivo. Of note, results from in vitro and in vivo screens on T cell-cancer cell interactions all point out to antigen presentation and interferon pathway alterations. An alternate conclusion would be to consider different interpretations of the in vitro screen results before making conclusions about their significance in treating patients with checkpoint inhibitor immunotherapies.

Minor comments:

In line 334, it would be better to state "To test" than "To prove". Someone could use a different model and prove the opposite to the author's conclusions.

Reviewer #2 (Remarks to the Author):

The authors have done an excellent job revising their manuscript and have addressed all of my comments.

Dear editors:

We appreciate the thoughtful examination of our revised manuscript, and are pleased that we addressed the majority of the reviewers' concerns. For this revised manuscript we have addressed the remaining comments from reviewer 1, as outlined in a point-by-point response below.

Reviewers' comments:

Reviewer #1 (Remarks to the Author):

The authors provide an improved resubmission addressing several of the comments from the initial review. There are additional issues to address:

Major comments:

The introduction still refers to “two melanoma patients” regarding deficiency of IFN γ R signaling pathway and resistance to PD-1 blockade therapy, despite the authors having added other relevant references with additional cases. The text should be equally corrected to match the new references.

We have revised the text to better encompass the additional references, and have removed the numerical designation. See page 3.

The B16.SIY is a very immunogenic variant of B16, with the inclusion of the SIY and DsRed foreign proteins. This is a model skewed towards a particular strong immune phenotype and the conclusions should reflect this point. The interpretation of the results is improved with the resubmission, but some of the conclusions continue to be too broadly stated to correctly reflect the data.

We agree that our conclusions apply to strongly but not weakly immunogenic tumors. In fact, this is one of the points of our results—one cannot have immune selection for a resistant variant unless there is an immune response to kill the more antigenic and susceptible tumor cells. We have added a reference from Dr. Robert Schreiber's work emphasizing this point (Matsushita et al., 2012). In fact, this is likely why poorly immunogenic B16 melanoma shows a lesser tumor outgrowth phenotype, and also a lesser therapeutic effect of checkpoint blockade (see additional discussion point below). Still, selection pressure over a longer time frame, maybe months or even years, might occur in humans, as the kinetics of tumor growth and immune response are likely distinct. We have added to the discussion to address the issue of immunogenicity. See page 15.

The studies using B16 in supplemental figure 3 are performed in a setting of a suboptimal number of implanted cells to maintain the author's argument that their data with B16.SIW.DsRed and MC38 is replicated in a lower immunogenicity model. However, this is not the case. This experiment should be preformed with a regular B16 cell inoculum and added to the main figures as opposed to being relegated to the supplemental materials. The text should reflect that it barely replicates the data with the highly immunogenic models used for the other studies.

We used 1×10^5 B16F10 tumor cells for two reasons. First, an inoculum of $1.0 \times 10^5 - 2.5 \times 10^5$ B16F10 cells is routinely used in the literature (Binnewies et al., 2019; Burr et al., 2017; Kaur et al., 2016; Pan et al., 2018; Rosato et al., 2019). Second, B16F10 is an abnormally aggressive tumor model. Lethality occurs much earlier compared to other tumor models, especially with a larger inoculum. Therefore, to give the immune system time to mount a response we used a

lower inoculum. If 2×10^6 tumor cells were used tumor growth would outpace the immune system and any immune-driven difference in tumor growth would be missed. We explicitly state our reasoning in the text and this approach was reported elsewhere (Juneja et al., 2017). In addition, as mentioned above, the lesser tumor outgrowth phenotype observed with WT B16 is expected because of diminished antigenicity, and we have amended the interpretation of these data with this perspective in mind within the text. See pages 7 and 15.

The text in page 9 continues to make the case that the author's data is at odds with findings in patients, but it is not as patients receive anti-PD-1 therapy and there is no such experiment in this work. Same comment is relevant for line 350.

We have changed the wording in the mentioned sections to better encompass our findings and hypotheses in the broader context of the literature surrounding acquired resistance. See pages 10 and 15.

The comment in lines 372-374 about the need to replicate data from in vitro screens with in vivo screens is fair, but skewed by the author's experience in interpreting their in vitro screen results in a way that did not anticipate a different result when tested in vivo. Of note, results from in vitro and in vivo screens on T cell-cancer cell interactions all point out to antigen presentation and interferon pathway alterations. An alternate conclusion would be to consider different interpretations of the in vitro screen results before making conclusions about their significance in treating patients with checkpoint inhibitor immunotherapies.

This is a fair comment and we have amended the Discussion to reflect this point. See page 14.

Minor comments:

In line 334, it would be better to state "To test" than "To prove". Someone could use a different model and prove the opposite to the author's conclusions.

We have made the suggested change. See page 11.

Reviewer #2 (Remarks to the Author):

The authors have done an excellent job revising their manuscript and have addressed all of my comments.

Binnewies, M., Mujal, A. M., Pollack, J. L., Combes, A. J., Hardison, E. A., Barry, K. C., et al. (2019). Unleashing Type-2 Dendritic Cells to Drive Protective Antitumor CD4+ T Cell Immunity. *Cell*, 177(3), 556–571.e16. <http://doi.org/10.1016/j.cell.2019.02.005>

Burr, M. L., Sparbier, C. E., Chan, Y.-C., Williamson, J. C., Woods, K., Beavis, P. A., et al. (2017). CMTM6 maintains the expression of PD-L1 and regulates anti-tumour immunity. *Nature*, 549(7670), 101–105. <http://doi.org/10.1038/nature23643>

Juneja, V. R., McGuire, K. A., Manguso, R. T., LaFleur, M. W., Collins, N., Haining, W. N., et al. (2017). PD-L1 on tumor cells is sufficient for immune evasion in immunogenic tumors and inhibits CD8 T cell cytotoxicity. *The Journal of Experimental Medicine*, 214(4), 895–904. <http://doi.org/10.1084/jem.20160801>

Kaur, A., Webster, M. R., Marchbank, K., Behera, R., Ndoeye, A., Kugel, C. H., et al. (2016). sFRP2 in the aged microenvironment drives melanoma metastasis and therapy resistance. *Nature*, 532(7598), 250–254. <http://doi.org/10.1038/nature17392>

- Matsushita, H., Vesely, M. D., Koboldt, D. C., Rickert, C. G., Uppaluri, R., Magrini, V. J., et al. (2012). Cancer exome analysis reveals a T-cell-dependent mechanism of cancer immunoediting. *Nature*, *482*(7385), 400–404. <http://doi.org/10.1038/nature10755>
- Pan, D., Kobayashi, A., Jiang, P., Ferrari de Andrade, L., Tay, R. E., Luoma, A. M., et al. (2018). A major chromatin regulator determines resistance of tumor cells to T cell-mediated killing. *Science*, *359*(6377), 770–775. <http://doi.org/10.1126/science.aao1710>
- Rosato, P. C., Wijeyesinghe, S., Stolley, J. M., Nelson, C. E., Davis, R. L., Manlove, L. S., et al. (2019). Virus-specific memory T cells populate tumors and can be repurposed for tumor immunotherapy. *Nature Communications*, *10*(1), 567. <http://doi.org/10.1038/s41467-019-08534-1>